



# Sunburned plankton: Ultraviolet radiation inhibition of phytoplankton photosynthesis in the Community Earth System Model version 2

Joshua Coupe[1,2], Nicole S. Lovenduski[1,2], Luise S. Gleason[3], Michael N. Levy[4], Kristen Krumhardt[4], Keith Lindsay[4], Charles Bardeen[5], Clay Tabor[6], Cheryl Harrison[7], Kenneth G. MacLeod[8], Siddhartha Mitra[9,10], and Julio Sepúlveda[2,11]

[1]Department of Atmospheric and Oceanic Sciences, University of Colorado, Boulder, CO, USA
[2]Institute of Arctic and Alpine Research, University of Colorado, Boulder, CO, USA
[3]Department of Earth Science, University of California Santa Barbara, Santa Barbara, CA, USA
[4]Climate and Global Dynamics Laboratory, NSF National Center for Atmospheric Research, Boulder, CO, USA
[5]National Center for Atmospheric Research, Boulder, CO, USA
[6]Department of Earth Sciences, University of Connecticut, Groton, CT, USA
[7]Department of Oceanography & Coastal Sciences, Louisiana State University, Baton Rouge, LA, USA
[8]Department of Geological Sciences, University of Missouri, Columbia, MO, USA
[9]Department of Geological Sciences, East Carolina University, Greenville, NC, USA
[10]Virginia Institute of Marine Science, Gloucester Point, VA, USA
[11]Department of Geological Sciences, University of Colorado, Boulder, CO, USA

**Correspondence:** Joshua Coupe (coupewx@gmail.com)





**Abstract.** Ultraviolet (UV) radiation can damage DNA and kill cells. We use laboratory and observational studies of the harmful effect of UV radiation on marine photosynthesizers to inform the implementation of a UV radiation damage function for phytoplankton photosynthesis in a modified version of the Community Earth System Model version 2 (CESM2-UVphyto). CESM2-UVphyto is capable of simulating UV inhibition of photosynthesis among modelled phytoplankton and ocean column

penetration of UV-A and UV-B radiation. We conduct a series of simulations with CESM2-UVphyto using the Marine Biogeo-chemistry Library (MARBL) ecosystem model to calibrate estimates of the sensitivity of phytoplankton productivity to UV radiation. Results indicate that increased UV radiation shifts the vertical distribution of phytoplankton biomass and productivity deeper into the column, causes a moderate decline in total global productivity, and changes phytoplankton community structure to favor diatoms. Our new CESM2-UVphyto model configuration can be used to quantify the potential ocean biogeochemical

and ecosystem impacts resulting from events that disturb the stratospheric ozone layer, such as an asteroid impact, a volcanic eruption, a nuclear war, and stratospheric aerosol injection-based geoengineering.

## 1    Introduction

Marine phytoplankton, unicellular photosynthesizing microorganisms, are responsible for almost half of the primary production on Earth and as a result comprise the base of the marine food web (Falkowski, 2012). Phytoplankton photosynthesis and the

associated drawdown of carbon dioxide are simulated in Earth system models with active ocean biogeochemistry and ecosystem components, which have been used to explore the response of global marine ecosystems to extreme events involving the injection of aerosols into the stratosphere (Lovenduski et al., 2020; Coupe et al., 2021; Harrison et al., 2022).

Photosynthesis requires light in a portion of the visible portion of the electromagnetic spectrum (400-700 nm wavelengths), i.e. photosynthetically active radiation (PAR). The relationship between phytoplankton photosynthesis and light is often rep-

resented by a photosynthesis-irradiance curve, where photosynthesis increases with light intensity until a specific threshold is reached at which point too much light can reduce phytoplankton photosynthetic capacity, also known as photoinhibition. This formulation implicitly includes ultraviolet (UV, 280-400nm) inhibition, but does not consider an increasing ratio of UV radiation to PAR. At high absolute levels of UV radiation, photosynthetic apparatuses are damaged faster than repairs can be made (Cullen et al., 1992; Smith and Cullen, 1995), especially when PAR is low relative to UV. While UV inhibition of

phytoplankton has been simulated in simple models (e.g., Cullen et al., 1992; Arrigo, 1994), most Earth system models do not explicitly include it.

Past work has used simple models to represent the biological effects of increased UV radiation on phytoplankton with biological weighting functions (BWF) that quantify the relative cellular damage caused by UV radiation as a function of wavelength for different phytoplankton functional types (PFTs). BWFs were determined from laboratory studies where phy-

toplankton cultures were exposed to varying amounts of UV radiation. Photoinhibition effects are calculated by integrating the BWF over UV wavelengths and incorporating damage by UV radiation along with photoadaptation, which may counteract UV damage. When UV-B radiation is high and PAR is low, adaptation opportunities can become overwhelmed by UV damage. While phytoplankton-specific BWFs have been implemented in simple models to capture the response to increased UV





radiation (e.g., Arrigo, 1994), no coupled Earth system models currently include representation of UV radiation inhibition in
marine phytoplankton. Yet, there are several potential use cases for Earth system models that incorporate UV inhibition of
phytoplankton photosynthesis.

Short-lived events that deplete stratospheric ozone, such as large asteroid impacts, volcanic eruptions, or even nuclear war,
may expose marine phytoplankton to harmful UV radiation by injecting aerosols and/or ozone depleting substances (ODS)
into the stratosphere. One of the most well known asteroid impacts caused an extinction event at the Cretaceous-Paleogene
boundary (K-Pg, ∼66 million years ago) when the 10 km diameter Chicxulub asteroid struck the shallow sea near the present
day Yucatan Peninsula (Alvarez et al., 1980; Smit and Hertogen, 1980; Schulte et al., 2010). Soot, dust, sulfur, carbon dioxide,
water vapor, among other gases were emitted high into the atmosphere and caused an impact winter. While reduced sunlight and
a sudden decline in temperature (Toon et al., 2016; Bardeen et al., 2017; Henehan et al., 2019; Tabor et al., 2020) were the most
likely main drivers of extinction of a significant amount of marine and terrestrial organisms (Jablonski et al., 1997; Henehan
et al., 2019; Tabor et al., 2020), the bombardment of the surface with UV radiation while life attempted to recover from the
impact winter may have slowed the return of photosynthesizers on land and in the ocean (Toon et al., 2016; Bardeen et al.,
2017, 2021). Direct evidence of UV exposure following asteroid impacts can be challenging to identify due to the destructive
nature of the cataclysmic events that may cause such anomalies. Paleoevidence of an increase in UV radiation contributing to
an extinction event can be found in the form of malformed plant spores at the Carboniferous-Devonian boundary, providing
some credibility to this mechanism (Marshall et al., 2020).

Model simulations allow for a quantification of ozone losses and associated increases in surface UV radiation. Simulations
of a global nuclear war support the hypothesis that surface UV radiation would increase after an injection of soot aerosols into
the stratosphere. The subsequent heating contributes to the decline of stratospheric ozone and increased surface UV-B radiation
(280 to 315 nm) in the 6 to 9 years after the war even as PAR remains below average (Bardeen et al., 2021). Similar to nuclear
war, asteroid impacts are likely to inject soot aerosols among dust, sulfates, and even halogens from vaporized seawater which
could deplete ozone beyond the effects of warmer stratospheric temperatures. Model simulations of asteroid impacts over water
show a decline in stratospheric ozone and increase in surface UV radiation, whether considering soot, dust, and water vapor
(Bardeen et al., 2021) or water vapor and halogens (Pierazzo et al., 2010).

Aside from past and future hypothetical extinction-level events, there are several examples in recent history of stratospheric
aerosol injection events disturbing the stratospheric ozone layer. Since 1970, several natural events have injected ODS into the
stratosphere and caused transient declines in stratospheric ozone. The 1982 eruption of El Chichón and the 1991 eruption of
Mt. Pinatubo released sulfur dioxide, water vapor, hydrochloric acid, and hydrobromic acid into the atmosphere (Evan et al.,
2023). After the Mt. Pinatubo eruption, these gases caused an increase in stratospheric sulfate aerosols and local warming of
the stratosphere, accelerated the heterogeneous chemical reactions that contribute to the destruction of ozone, and slowed the
transport of ozone rich air from the tropics, causing up to a 10% reduction in stratospheric ozone in the Northern Hemisphere
mid-latitudes (Østerstrøm et al., 2023). In 2021, the Hunga Tonga submarine volcano eruption caused a 5% decline in strato-
spheric ozone in the tropics after it injected a large amount of water vapor into the stratosphere (∼50 Tg) in addition to sulfates
(Vömel et al., 2022; Evan et al., 2023; Zhu et al., 2023). While most volcanic eruptions inject a plume of predominantly sulfur





dioxide in the stratosphere, this anomalous water vapor injection demonstrated the vulnerability of stratospheric ozone to a
wide variety of emissions across a range of chemical species. Increased surface UV radiation from recent or potential future
volcanic eruptions has been studied but possible ecosystem impacts as a result have not yet been studied in detail. Laboratory
results of the estimated impact of ultraviolet radiation on phytoplankton shows there exists the potential for the ozone loss sim-
ulated in models of these volcanic eruptions to inhibit phytoplankton growth (Cullen et al., 1992). Similarly, the consequence
of anthropogenic stratospheric aerosol emissions and its impacts on ocean ecosystems are understudied.

Several "geoengineering" or climate intervention strategies have been proposed to either slow or reverse global warming,
but the evaluation of such actions aimed at the mitigation of climate hazards would benefit from better understanding their
potential consequences for stratospheric ozone (Stratospheric aerosol injection; MacMartin et al., 2019; Tilmes et al., 2020).
Deliberate injections of scattering aerosols into the stratosphere, which mimic the global cooling following volcanic eruptions
that release sulfur, are likely to cause stratospheric warming and increase the heterogeneous chemical reactions that produce a
depleted stratospheric ozone state (Tilmes et al., 2022). Deliberate sulfur based particle injections to reduce global mean surface
temperatures were found to deepen the Antarctic ozone hole within 10 years using a high-top version of the Community Earth
System Model (CESM) with interactive stratospheric chemistry (Tilmes et al., 2021). While little is known about the biological
impacts of these climate intervention strategies, we can look to the past for an example of anthropogenic emission of an ODS
that has been studied for its effects on marine ecosystems.

The multi-decadal increase in atmospheric chlorofluorocarbon (CFC) concentrations was responsible for the formation of
the Antarctic ozone hole (Solomon et al., 1986), which increased surface UV radiation as far north as Australia and spurred
significant scientific research into the biological effects of UV radiation. The global ban on manufacturing of CFCs via the
Montreal Protocol in 1987 halted emissions to near zero, enabling the recovery of the ozone hole in the following decades
(Garcia et al., 2012). Though this destruction and recovery of stratospheric ozone likely had significant biological impacts, the
lack of observational studies documenting temporal trends among marine phytoplankton limits scientists' ability to understand
the biological response of this community as a function of changing UV irradiance (Smith and Cullen, 1995). In lieu of
observational studies, lab grown phytoplankton can be exposed to varying levels of UV radiation to quantify their sensitivity
to something like an ozone hole (Cullen et al., 1992).

Finally, marine phytoplankton exposure to UV radiation may increase as anthropogenic climate change warms the Earth's
surface, representing a compounding threat. As carbon dioxide emissions continue to warm the Earth's surface, the density
gradient in the upper ocean increases (Li et al., 2020), resulting in increased stratification that concentrates phytoplankton in
the surface layers of the ocean where these organisms are exposed to higher amounts of harmful UV radiation (Gao et al.,
2019). Mineralizing phytoplankton, such as diatoms (silica frustules) and coccolithophores (calcium carbonate shells) rely on
their shell-like structures for physical protection, but the production of these shells may be disrupted by UV radiation (Neale
et al., 1998; Lorenzo et al., 2019). Coccolithophore shells, also known as coccospheres, play a role in filtering damaging UV
light (Xu et al., 2016), but may thin in response to ocean acidification (Ridgwell et al., 2009; Fox et al., 2020; Krumhardt
et al., 2019). Diatom frustules offer similar protection from UV radiation, but in contrast to coccospheres, these silica-based
structures dissolve more slowly in an acidified ocean (Aguirre et al., 2018; Taucher et al., 2022).





Numerous potential use cases motivate our development of an Earth system model capable of simulating phytoplankton
photosynthesis inhibition by UV radiation. Here, we describe the implementation of this new capability in the Community
Earth System Model version 2. We test the new model formulation using simulations with varying levels of UV radiation. As
we will demonstrate, UV radiation perturbs the vertical distribution of phytoplankton biomass and productivity while causing
moderate decline in global phytoplankton productivity and changes in phytoplankton community structure. Our new model
addition is a useful tool for exploring both natural and anthropogenic events that may increase marine phytoplankton exposure
to UV radiation.

## 2 Materials and Methods

### 2.1 Model Components

We use the Community Earth System Model version 2.1.5 (CESM2) with atmosphere, ocean, sea ice, and land compo-
nents that exchange information with each other through a coupler (Danabasoglu et al., 2020). Our CESM2 code modifications
produce a model that is capable of simulating UV-B (280 to 315 nm) and UV-A (315 to 400 nm) radiation in the atmosphere,
propagate it through the ocean, and simulate UV inhibition of phytoplankton photosynthesis. We call our modified model
version CESM2-UVphyto.

The atmospheric model used for this study is the Whole Atmosphere Community Climate model version 4 (WACCM4,
Marsh et al., 2013), using a grid with 1.9 x 2.5 degree horizontal resolution, 66 vertical layers and a 140 km model top.
WACCM4 model includes a mathematical representation of stratospheric circulation, thermodynamics, and chemistry, the last
of which is crucial for simulating UV radiation. As a "high top" model it can resolve the stratosphere in addition to parts
of the mesosphere. The Tropospheric Ultraviolet and Visible (TUV) model version 4.2 (Zerefos and Bais, 1997) is added to
WACCM4, as Bardeen et al. (2021) did, and calculates spectral integrals in-line across 100 wavelength intervals between 120
and 750 nm, instead of using a look-up table approach. Radiation between 280 nm and 400 nm from TUV is used to compute
biologically relevant parameters. The Rapid Radiative Transfer Model for GCMs (RRTMG; Iacono et al., 2000) is used for
atmospheric radiative transfer not within the UV range.

WACCM4 includes the chemistry modifications described in Bardeen et al. (2021) and a module to inject stratospheric soot
and ozone-depleting halogens, specifically hydrogen bromide and hydrogen chloride. TUV is modified to allow actinic flux to
be affected by the optical effects of aerosols. Biological weighting functions that determine UV inhibition of phytoplankton
photosynthesis are also incorporated into TUV (Zerefos and Bais, 1997; Bardeen et al., 2021), which also calculates surface
UV-A, UV-B, and some UV-C radiation (model computes spectral integrals over 121 nm to 280 nm, compared to the defined
UV-C range of 100 nm to 280 nm). The spectrally integrated biological weighting functions are computed explicitly over UV-B
and UV-A radiation and sent to the coupler and then to the ocean at hourly frequency, identical to the treatment of shortwave
radiation by RRTMG.

The ocean model component is the Parallel Ocean Program version 2 (POP2; Danabasoglu et al., 2020). The ocean model
has a nominal horizontal resolution of 1 degree with 60 vertical levels, with a uniform vertical resolution of 10 m in the upper



150 m, which increases to a maximum vertical spacing of 250 m between 3500 m and its maximum depth of 5500 m. POP2 is coupled to an ocean biogechemistry and ecosystem model called the Marine Biogeochemistry Library (MARBL), which has the flexibility to resolve different plankton configurations (MARBL; Long et al., 2021). The version of MARBL used here

includes four phytoplankton functional types (PFTs), small phytoplankton, diatoms, diazotrophs, and coccolithophores, and two zooplankton functional types (microzooplankton and mesozooplankton), a configuration referred to as "4p2z" (Krumhardt et al., 2024). MARBL-4p2z simulates multiple nutrient co-limitation for all four PFTs.

In MARBL, small phytoplankton are limited by nitrogen, phosphorus, and iron. The small phytoplankton functional group represents a diverse group of phytoplankton, from *Prochlorococcus* and *Synechococcus* in warm, oligotrophic regions to groups

like *Phaeocystis*, cryptophytes, and picoeukaryotes in upwelling or polar regions. Diazotrophs are nitrogen fixing bacteria that are limited by phosphorus and iron. Diatoms are silicifiers that are limited by nitrogen, phosphorus, iron, and are the only PFT limited by silicon. Coccolithophores are limited by nitrogen, phosphorus, iron, and are the only PFT limited by carbon. The coccolithophore PFT is primarily represented as *Emiliana huxleyi* in MARBL-4p2z, based on Krumhardt et al. (2019), and simulates calcium carbonate production as a function of temperature, aqueous $CO_2$, and phosphorus limitation. Micro-

zooplankton graze on phytoplankton, while mesozooplankton graze on both phytoplankton and microzooplankton. Other than grazing, loss of phytoplankton can occur through aggregation and subsequent sinking through the column. We also performed UV sensitivity simulations using two other configurations of MARBL: "4p1z" (4 phytoplankton and only 1 zooplankton), and "3p1z" (3 phytoplankton and 1 zooplankton). In the 3p1z configuration, coccolithophores are not represented but an implicit calcifier is included as a changing percentage of the small phytoplankton PFT.

Phytoplankton photosynthesis in all MARBL configurations is calculated by scaling the maximum growth rate by limitation functions for temperature, light, and nutrient availability (Long et al., 2021). Phytoplankton photosynthesis increases in response to increasing PAR, following the light limitation formulation from Geider et al. (1998). Photoadaptation is represented as a varying chlorophyll to carbon ratio for each PFT, depending on light, temperature, and nutrient limitations (Geider et al., 1998). PAR in MARBL is estimated as 45% of surface shortwave radiation and is attenuated with depth as a func-

tion of chlorophyll concentration, taking into consideration shading by phytoplankton. Zooplankton are not considered in the light attenuation equation, as they do not contain chlorophyll and do not absorb light as effectively as phytoplankton. In high concentrations, zooplankton may impact light attenuation, but these populations tend to overlap with high phytoplankton concentrations which already significantly attenuate light. In CESM2-UVphyto, phytoplankton growth rates can be slowed by increased UV radiation, as described in Section 2.2. While TUV provides a direct calculation of PAR, the simulations here

use the PAR estimation from shortwave radiation in order to test the sensitivity of the model to the new UV radiation scheme alone.

Source code modifications to WACCM4, TUV, POP2, and MARBL were required to enable the calculation of UV inhibition. In WACCM4, this includes the integration of TUV and modules used to simulate aerosol injections relevant for asteroid impacts (Bardeen et al., 2017, 2021). New BWFs relevant for the phytoplankton species simulated in MARBL were added to TUV.

Modifications to the CESM2 model coupler were made to pass UV radiation fields to POP2, and POP2 was modified to



pass information to MARBL, where UV inhibition of photosynthesis is calculated. All other components are used without modification from the cesm2.1.5 code base.

The sea ice model is CICE5 with interactive sea ice that shares the same horizontal grid as the ocean model (Hunke et al., 2015). CICE5 includes mushy-layer thermodynamics and a melt pond parameterization that produces melt ponds preferentially
on undeformed sea ice. Sea ice is allowed to affect PAR attenuation with depth in the ocean model using a subgrid-scale sea ice thickness distribution (Long et al., 2015). Essentially, various sea ice thickness categories can impact the ability of phytoplankton to grow underneath sea ice. In CESM2-UVphyo, UV radiation interaction with sea ice is identical to the treatment of shortwave radiation, until UV radiation reaches the water, at which point its attenuation accelerates rapidly compared to PAR.

The land model is the Community Land Model version 5 (CLM5) with a carbon-nitrogen cycle (Lawrence et al., 2019). It
simulates the evolution of the land physical state, characteristics of the land surface, exchanges of energy and material with the atmosphere, and run-off into the ocean. CLM5 has a horizontal resolution of 1.9 x 2.5 degrees which is shared with the atmosphere grid, with 15 vertical layers for the land and 10 vertical layers for lakes. Photosynthesis of terrestrial vegetation is not inhibited by UV radiation, but future work is expected to incorporate this functionality.

### 2.2 Calculation of Ultraviolet Inhibition of Photosynthesis

Biological weighting functions (BWF) for UV inhibition for each PFT (see Section 2.3) are integrated over the UV-A and UV-B portion of the electromagnetic spectrum in TUV and used to calculate surface $E^*_{inh}$, a biologically weighted dimensionless UV dosage rate, according to Cullen et al. (1992),

$$E^*_{inh} = \sum_{\lambda=280nm}^{400} E(\lambda) \cdot \epsilon(\lambda) \cdot \Delta\lambda, \tag{1}$$

where $E(\lambda)$ is the surface spectral irradiance at wavelength $\lambda$, and $\epsilon(\lambda)$ is the BWF ($(\text{mW m}^2)^{-1}$) representing the inhibition
of photosynthesis by UV radiation. $E^*_{inh}$ can be thought of as the total integrated potential damage phytoplankton will suffer from across radiation in the UV spectrum. The BWF, given by $\epsilon(\lambda)$, describes how each wavelength interval contributes to that damage. $E^*_{inh}$ values are determined at every surface model grid cell and time step for small phytoplankton, diatoms, diazotrophs, and coccolithophores.

$E^*_{inh}$ is passed from the atmospheric model (TUV) to the marine biogeochemistry model (MARBL) and propagated verti-
cally through the water column. To capture the rapid attenuation of UV radiation with depth in the ocean, we compute $E^*_{inh}$ at a given model level using the equation in Overmans and Agustí (2020), which relates depth profiles of chlorophyll concentration and UV-B radiation ($\lambda = 305$ nm) for oligotrophic ocean conditions:

$$K_{d-UV} = 0.14 \cdot \text{CHL} + 0.29, \tag{2}$$

where the attenuation coefficient, $K_{d-UV}$, is computed as a function of chlorophyll (CHL, mg m$^{-3}$). This equation was
determined empirically using stationary radiometers collecting spectrally integrated UV-B (280 to 315 nm) and UV-A (315 to





400 nm) radiation over 10 minute intervals along coral reefs in the Red Sea and measurements of chlorophyll-a using a Trilogy Fluorometer (Overmans and Agustí, 2020). $E_{inh}^*$ is propagated through the column as a function of $K_{d-UV}$ and the thickness of a particular model layer, $dz$ (m):

$$E_{inh}^*(k+1) = E_{inh}^*(k) \cdot exp(-K_{d-UV} \cdot dz), \tag{3}$$

where $k$ is the vertical layer index. According to this relationship, for chlorophyll concentrations of 0.02 mg m$^{-3}$ (the minimum value used for calculating the impact of chlorophyll on the vertical propagation of PAR in MARBL), the depth at which UV radiation declines to 1% of the surface value is ∼16 m. At higher chlorophyll concentrations (∼0.85 mg m$^{-3}$, approximately the highest peak found in Overmans and Agustí (2020)) the depth at which surface UV radiation attenuates to 1% is ∼11 m.

UV inhibition of photosynthesis in CESM2-UVphyto primarily occurs in the upper 20 m of the water column and depends
greatly on shading by phytoplankton. Because POP has upper layers that are 10 m thick, it is only the top two layers of the ocean that experience UV inhibition. We use the mid-point of the layer to calculate attenuation for each layer, which may underestimate the influence of much higher $E_{inh}^*$ within the top 5 m of the level, but could overestimate $E_{inh}^*$ within the bottom 5 m of the level. The effect of ocean surface roughness from waves on UV radiation entering the surface ocean is not considered in the attenuation of UV radiation.

Finally, a UV inhibition term ($\gamma_{UV}$) is computed at every model level, similar to nutrient and light limitation, using the functional form from Arrigo (1994):

$$\gamma_{UV} = \frac{1}{1 + E_{inh}^*}, \tag{4}$$

where $\gamma_{UV}$ ranges from 0 (total UV inhibition) to 1 (no UV inhibition) and is included in the calculation of the growth rate for phytoplankton from Long et al. (2021),

$$\mu_i = \mu_{max}(T) \cdot \gamma_N \cdot \gamma_l \cdot \gamma_{UV}, \tag{5}$$

where $\mu_i$ represents the growth rate of a particular PFT, $\mu_{max}(T)$ represents the maximum growth rate at a given temperature, and $\gamma_N$ and $\gamma_l$ represent nutrient and light limitation terms, respectively, ranging from 0 (no nutrients/light) to 1 (sufficient nutrients/light). In this way the $\gamma$ limitation terms reduce the maximum growth rate multiplicatively.

### 2.3 Determination of Biological Weighting Functions

The expansion of the Southern Hemisphere ozone hole during the 1980s due to the emission of chlorofluorocarbons in the decades prior prompted studies into the susceptibility of marine life and phytoplankton to UV radiation. These efforts produced biological weighting functions (BWFs) specific to different PFTs that are suitable for the simulation of UV inhibition in photosynthesis. We utilized both laboratory and *in-situ* studies with appropriate BWFs for the phytoplankton community



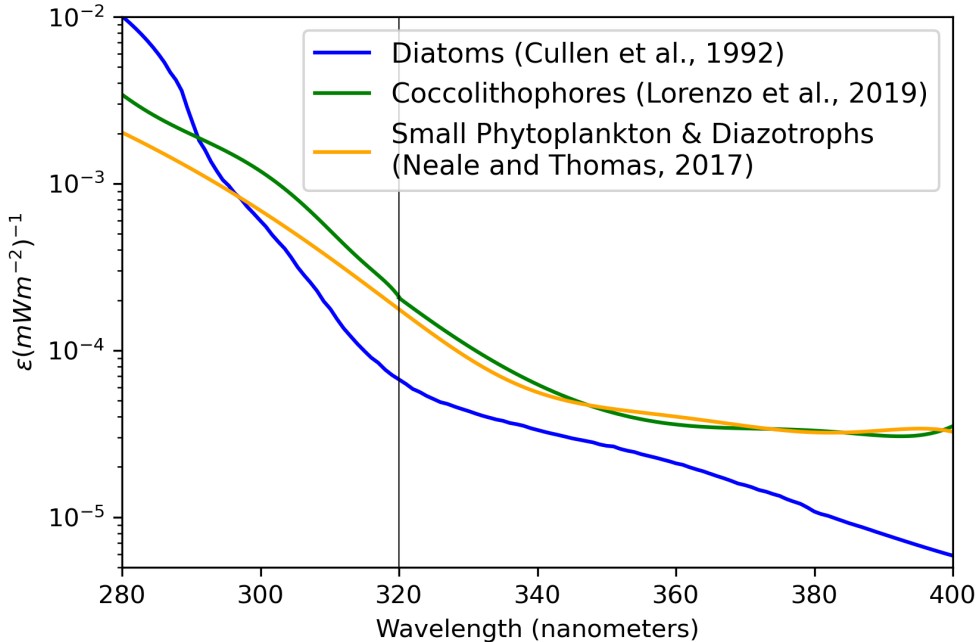

**Figure 1.** Species-specific biological weighting functions showing the biological efficiency for damage to phytoplankton photosynthesis by ultraviolet light as a function of wavelength. Depicted here are the weighting functions for diatoms (Cullen et al., 1992), coccolithophores (Lorenzo et al., 2019), small phytoplankton and diazotrophs (Neale and Thomas, 2017) that were used in our simulations of CESM2-UV. Black line represents the division between UV-A and UV-B radiation.

represented within MARBL (Figure 1). For diatoms, we used the BWF reported in Cullen et al. (1992), who measured changes

in productivity of the diatom *Phaeodactylum sp.* by exposing samples of the species to a range of wavelengths of UV and PAR. A look-up table is provided in Supplemental Table 1. Wavelengths were interpolated to the bounds provided in this table to calculate spectral integrals. Diatoms show relatively low sensitivity to UV-A light, but are exponentially more sensitive to UV-B light, especially for wavelengths shorter than 290 nm.

   For coccolithophores, which are designed to represent the species *Emiliana huxleyi* in MARBL, we adopted the BWF

reported by Lorenzo et al. (2019)—a $6^{th}$ order polynomial that separately computes $\epsilon(\lambda)$ for UV-B and UV-A wavelengths in Table 1. Lorenzo et al. (2019) used similar methods for determining sensitivity to UV radiation as Cullen et al. (1992), but also explored how elevated atmospheric $CO_2$ concentrations impacted the sensitivity of coccolithophore photosynthesis to UV radiation. At elevated $CO_2$ concentrations, *Emiliana huxleyi* cells exhibit decreased rates of calcification (Lorenzo et al., 2019). Affected cells may be able to reallocate energy towards producing organic material despite difficulties in forming calcium

carbonate shell material. Some work has suggested that thinner coccospheres will allow more UV radiation to penetrate into the cell, making the cell more susceptible to photoinhibition (Guan and Gao, 2010). This sensitivity is explored in Section 2.4.

**Table 1.** Different phytoplankton functional types (PFTs) and the study that determined the specific biological weighting functions (BWFs) we used in simulations of CESM2-UVphyto. Units are in $(\text{mW m}^{-2})^{-1}$.

| Phytoplankton | Study | Biological Weighting Functions |
|---|---|---|
| Siliceous Diatoms | Cullen et al. (1992) | Provided in look-up table (see Table S1). |
| Calcifying Coccolithophores | Lorenzo et al. (2019) | $\epsilon(\lambda)$ for UV-B $= (-1.40530706\text{x}10^{-12})\lambda^6 + (2.39353416\text{x}10^{-9})\lambda^5 - (1.68928198\text{x}10^{-6})\lambda^4 + (6.31865068\text{x}10^{-4})\lambda^3 - 0.131975456\lambda^2 + 14.5753274\lambda - 663.797953$ <br> $\epsilon(\lambda)$ for UV-A $= (8.89581784\text{x}10^{-15})\lambda^6 - (1.93953328\text{x}10^{-11})\lambda^5 + (1.76199028\text{x}10^{-8})\lambda^4 - (8.53798419\text{x}10^{-6})\lambda^3 + 0.00232766669\lambda^2 - 0.338550975\lambda + 20.5261692$ |
| Small phytoplankton and Diazotrophs | Neale and Thomas (2017) | $\epsilon(\lambda) = (-3.10572006\text{x}10^{-15})\lambda^6 + (6.27284524\text{x}10^{-12})\lambda^5 - (5.22236845\text{x}10^{-9})\lambda^4 + (2.28848111\text{x}10^{-6})\lambda^3 - (5.54843743\text{x}10^{-4})\lambda^2 + 0.0702209062\lambda - 3.59639531$ |

We use the BWF reported in Neale and Thomas (2017) for both small phytoplankton and diazotrophs. Neale and Thomas (2017) studied UV sensitivity of *Prochlorococcus* and *Synechococcus* from temperate and tropical open-ocean regions under different UV exposures and water temperatures. The BWF from Neale and Thomas (2017) closely resembles the species for

small phytoplankton in MARBL and is based on a 7-28% decline in productivity due to enhanced UV radiation in a laboratory study.

The effects of UV radiation on diazotrophs has been examined to some extent (Lesser, 2008; Cai et al., 2017), but none have reported usable BWFs for computing UV damage as a function of wavelength. Lesser (2008) found that diazotrophs were quite sensitive to UV radiation, while Cai et al. (2017) found that UV-absorbing compounds in high-light grown diazotrophs

(*Trichodesmium*) can mitigate some but not all UV damage. The shallow depth of *Trichodesmium* coupled with its importance of supplying nitrogen to oligotrophic gyre ecosystems suggests a potentially important regional role for UV inhibition of nitrogen fixers. We generalize diazotrophs by using the same BWF as reported by Neale and Thomas (2017), therefore representing a mid-point between the BWFs for the diatoms and coccolithophores. A $6^{th}$ order polynomial to compute $\epsilon(\lambda)$ for all UV wavelengths for small phytoplankton and diazotrophs is provided in Table 1. Diazotrophs lack mineral shells like diatoms and

coccolithophores, and thus are more akin to small phytoplankton in their sensitivity to UV. Diazotrophs comprise less than 3% of global phytoplankton net primary productivity in pre-industrial MARBL-3p1z simulations, meaning large uncertainties in how diazotrophs will respond to UV radiation do not significantly affect global productivity or nutrient cycling over shorter (decadal) timescales but may have regional impacts that CESM2-UVphyto can consider.





## 2.4 Scaling UV inhibition of coccolithophore photosynthesis as a function of shell thickness

Coccolithophores construct shells out of calcium carbonate, providing protection from grazing, viral attacks, as well as from harmful amounts of PAR and UV radiation (Monteiro et al., 2016; Krumhardt et al., 2019; Xu et al., 2016; Lorenzo et al., 2019). In MARBL, coccolithophores have the lowest maximum grazing rate, partly because of the protection from their shell. Laboratory studies of calcified and thin-shelled, or "naked" coccolithophores have found that the shell reduces the transmission of UV-B radiation by 18% (Xu et al., 2016). In response to decreased transmission of UV-B, calcified cells in outdoor conditions

were also found to have 3.5 times higher growth rates than "naked" coccolithophores (Xu et al., 2016).

In CESM2-UVphyto, coccolithophore shell thickness is related to the instantaneous ratio of particulate inorganic carbon (PIC; calcium carbonate) to particulate organic carbon (POC), $\frac{PIC}{POC}$ (Krumhardt et al., 2019). Shelled coccolithophores have a $\frac{PIC}{POC} > 0.05$, while 'naked' coccolithophores have a $\frac{PIC}{POC} <= 0.05$. To accommodate the UV inhibition of coccolithophore photosynthesis that accompanies thinning coccolithophore shells, we modify $\gamma_{UV}$ when sufficient UV radiation is present

($E_{inh}^* > 0.5$) as follows:

$$\gamma_{UV}(\text{shelled}) = C \cdot \gamma_{UV}(\text{naked}), \tag{6}$$

where

$$C = \begin{cases} 3.5^{-1}, & \text{if } \frac{PIC}{POC} < 0.05, \\ \left(\frac{PIC}{POC}\right)^{\frac{1}{6}}, & \text{otherwise.} \end{cases} \tag{7}$$

Our $\gamma_{UV}$ scaling aligns with the Xu et al. (2016) finding that calcified cells have growth rates 3.5 times higher than naked shells

under moderate levels of UV radiation. Results from simulations without PIC/POC scaling of UV inhibition to coccolithophores are shown in the Supplemental Information document.

## 2.5 Pre-industrial simulations with low UV radiation

We conduct two simulations with CESM2-UVphyto under no or low levels of UV radiation to ensure that our new BWFs do not disrupt the normal functioning of the biogeochemical and ecosystem model. Both of these test simulations use the

4p2z configuration of MARBL and pre-industrial atmospheric forcing consistent with the year 1850 and atmospheric $CO_2$ concentration set to 284.7 ppm. The first is a 10-year coupled simulation of CESM2-UVphyto with an $E_{inh}^*$ value of zero that branches from a previously spun-up ocean physical and biogeochemical state Krumhardt et al. (2024); the UV forcing from this simulation is referred to as $E_{inh}^*(0)$. The end of the $E_{inh}^*(0)$ simulation is used as the initial condition for all subsequent simulations. The second simulation probes the impact of the new BWFs on the resulting biogeochemical state under 5 years of

pre-industrial (*i.e.*, low) UV radiation levels, $E_{inh}^*(PI)$. This simulation of ocean biogeochemical sensitivity to pre-industrial levels of UV radiation (and subsequent ones) are conducted in ocean-ice only mode, *i.e.*, a simulation of CESM2-UVphyto





with only the ocean and ice component models that is forced by saved surface states and fluxes from coupler output files; UV-A, UV-B, and UV-C radiation; and $E_{inh}^*$ for each BWF at the air-sea interface. These forcings are derived from a year-long, fully coupled simulation of CESM2-UVphyto with WACCM4 coupled to TUV under pre-industrial levels of UV radiation, and produce a global average annual mean surface $E_{inh}^*$(diatoms) of 0.31, $E_{inh}^*$(small phytoplankton & diazotrophs) of 0.71, and $E_{inh}^*$(coccolithophores) of 0.74. The surface UV radiation levels present in these simulations are well within the range of those used in laboratory studies that informed the construction of BWFs. We note that while a single year of forcing does not capture interannual variability and could potentially be anomalous, we see below that the results from the forced $E_{inh}^*(PI)$ simulation are indistinguishable from the coupled $E_{inh}^*(0)$ simulation.

### 2.6 Pre-industrial simulations with elevated UV radiation

We conduct 5 year simulations to explore the modeled biogeochemical and ecological response to extremely high levels of surface UV radiation. In a case referred to as $E_{inh}^*(20x\ PI)$, forcings at the air-sea interface are derived from previously described year-long, fully coupled simulation of CESM2-UVphyto with WACCM4-TUV, except for $E_{inh}^*$. $E_{inh}^*(20x\ PI)$ uses coupler fluxes from the $E_{inh}^*(PI)$ simulation with only $E_{inh}^*$ increased by a factor of 20 for all PFTs. This value is approximately the average $E_{inh}^*$ response for a large injection of soot, hydrogen bromide and hydrogen chloride into the stratosphere, destroying 95% of all stratospheric ozone globally. This UV forcing is not intended to perfectly simulate an asteroid impact, but represents an upper bound of possible surface UV radiation to test the upper range of the selected BWFs. The simulation is run for five years, or five cycles of the year-long saved forcing.

### 2.7 Elevated CO$_2$ simulations

At higher atmospheric $CO_2$ concentrations, coccolithophores in some regions may experience a carbon fertilization effect as carbon used for photosynthesis become less limiting, allowing for faster growth compared to lower $CO_2$ concentrations (Krumhardt et al., 2017, 2019). However, coccolithophores' ability to construct and maintain thick calcium carbonate shells (i.e., high PIC/POC ratios) may be inhibited (Krumhardt et al., 2019). To explore UV inhibition in conditions that engender thinner coccolithophore shells, we conducted a range of ocean and sea-ice only simulations with varying atmospheric $CO_2$ concentrations at the ocean's surface: 284 ppm, 400 ppm, 600 ppm, 700 ppm, and 900 ppm. First, these simulations are spun-up with no UV radiation inhibition ( $E_{inh}^*(0)$ ) and $CO_2$= 284.7 ppm until the upper ocean (0-100m) global average pH exhibits little change with time, occurring after 15 years of simulation. The simulation with $CO_2$= 400 ppm uses the end of the $CO_2$ = 284 ppm simulation as an initial condition, the end of the simulation with $CO_2$= 400 ppm is used as the initial condition for the $CO_2$= 600 ppm simulation, the end of the simulation with $CO_2$=600 ppm is used to start the $CO_2$=700 ppm simulation, and the end of the simulation with $CO_2$=700 ppm is used to start the $CO_2$=900 ppm simulation. This approach speeds up the spin-up as atmospheric $CO_2$ concentrations progress higher. All simulations with $CO_2$ levels greater than 284.7 ppm are run for 17 years, at which point variations in upper ocean global average pH are smaller than 0.01 from year to year. Next, $E_{inh}^*$(PI) is imposed for 1 year starting at the end of each simulation with incrementally increasing atmospheric $CO_2$ (400 ppm, 600 ppm, 700 ppm, 900 ppm). Finally, $E_{inh}^*(20x\ PI)$ is imposed for 1 year in the same manner as $E_{inh}^*$(PI) , using the same initial condition.



## 3 Results

UV radiation at the ocean's surface causes a reduction in globally integrated phytoplankton net primary productivity (NPP) through inhibition of photosynthesis in the top 20 m of the ocean under $E^*_{inh}$(PI) and $E^*_{inh}$(20x PI) forcing. Under the control $E^*_{inh}(0)$ forcing, annual-mean globally integrated top 150 m NPP equilibrates to 55 Pg C yr$^{-1}$ in the 4p2z configuration, which is well within the range of satellite estimates (43-67 Pg C yr$^{-1}$ ; Behrenfeld and Falkowski, 1997 and Behrenfeld et al., 2006) and similar to other configurations of MARBL (see Supplemental Information: Phytoplankton Validation). Under $E^*_{inh}$(PI) UV forcing, annual-mean global NPP is reduced by 1% compared to $E^*_{inh}(0)$, as shown in Figure 2a. Our findings are consistent with the hypothesis that pre-industrial UV radiation should not significantly impact globally integrated NPP. In the $E^*_{inh}$(20x PI) case, annual-mean globally integrated NPP declines by 7.5% to 50.8 Pg C yr$^{-1}$ compared to the $E^*_{inh}(0)$ simulation (Figure 2a). The global NPP response to larger UV forcing in the $E^*_{inh}$(20x PI) case is within the very large range of anticipated response of phytoplankton to UV inhibition in the context of increased UV radiation under the Southern Hemisphere ozone hole (see Supplemental Information: Phytoplankton Under Ozone Hole Stress).

The phytoplankton productivity response to UV inhibition is a function of characteristics unique to each phytoplankton type, with some types showing increases in productivity at the expense of others. Small phytoplankton constitute half of the globally integrated NPP in both the $E^*_{inh}(0)$ and $E^*_{inh}$(PI) simulations (Figure 2b), while diatoms contribute approximately 40%. Both coccolithophores and diazotrophs combined make up around 10%. Under preindustrial levels of UV radiation (case $E^*_{inh}$(PI)), the relative contribution of each PFT to global NPP is virtually unchanged and the timing of seasonal blooms are not significantly altered when compared to the $E^*_{inh}(0)$ case (Figure 2b). Under $E^*_{inh}$(20x PI) forcing, diatom productivity increases, and small phytoplankton productivity decreases relative to the $E^*_{inh}(0)$ case (Figure 2b). During the month of April, small phytoplankton shift from comprising a clear majority of global phytoplankton NPP to generating nearly the same global productivity as diatoms in the $E^*_{inh}$(20x PI) case (Figure 2b). The relative contribution of diazotrophs to global total NPP is not modified with UV radiation (Figure 2)b. Enhanced UV radiation reduces the contribution of coccolithophore productivity to global NPP from December to February.

The relative responses of phytoplankton NPP to UV radiation across the different PFTs are driven primarily by the BWFs of the different PFTs (Figure 1) which generate $E^*_{inh}$ (Figure 3). In the $E^*_{inh}$(20x PI) simulation, coccolithophore UV damage ($E^*_{inh}$) is the largest of all the PFTs in all months of the year, followed by small phytoplankton/diazotroph and then diatom damage (Figure 3a). This finding is consistent with the relative UV damage ($E^*_{inh}$) across the PFTs in the $E^*_{inh}$(PI) simulation (Figure 3a), even though the weighting functions of the PFTs exhibit different responses to UV radiation as a function of wavelength (Figure 1). One could imagine conducting a simulation with CESM2-UVphyto forced by a large aerosol injection that depletes stratospheric ozone in ways that non-linearly increase UV-B radiation at wavelengths between 280 and 290 nm, for which diatom UV damage would be larger than those of the other PFTs (see, e.g., Figure 1), which would produce a very different NPP response. The net impact of UV radiation on phytoplankton NPP in CESM2-UVphyto is further determined by the latitude and depth at which the majority of the population of each PFT resides. For example, the highest $E^*_{inh}$ values can be found at the lowest latitudes (Figure 3b) and small phytoplankton exist in greatest numbers at low latitudes, indicating

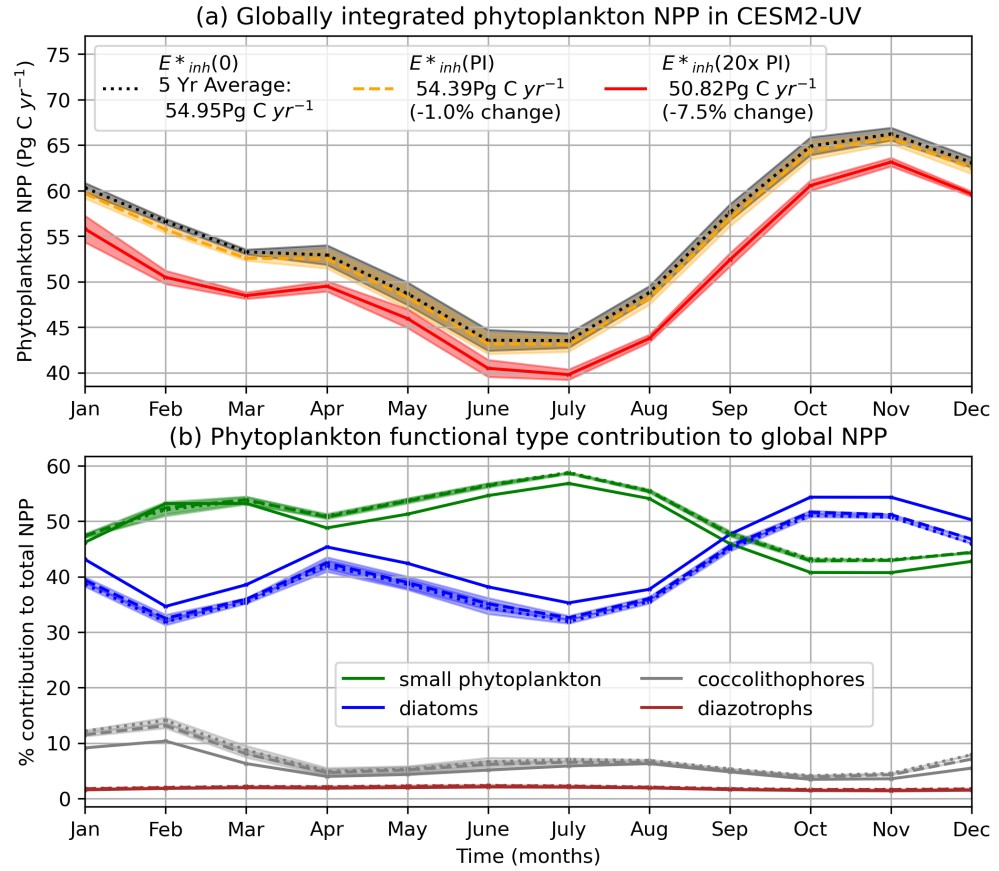

**Figure 2.** (a) Globally integrated 5-year mean monthly NPP climatology integrated over the top 150 m (Pg C $yr^{-1}$) in the $E^*_{inh}(0)$, $E*_{inh}$(PI), and $E*_{inh}$(20x PI) simulations. Shading indicates 1 standard deviation below and above the 5-year mean for each month. (b) Each phytoplankton functional type's % contribution to globally-integrated NPP in the (solid) $E^*_{inh}$(20x PI), (dashed) $E^*_{inh}$(PI), and (dotted) $E^*_{inh}(0)$ simulations. Note the $E^*_{inh}$(PI) and $E^*_{inh}(0)$ simulations are nearly indistinguishable.

high vulnerability relative to the other phytoplankton types. Under preindustrial levels of UV radiation (case $E^*_{inh}$(PI)), the
spatial distribution of phytoplankton productivity for each PFT (Figure 4a) is similar to the distributions reported in Long et al. (2021) using a MARBL-3p1z configuration that resolves these functional types. Coccolithophore NPP (Figure 4a) and their $CaCO_3$ production (not shown) share a similar spatial pattern with that reported in Krumhardt et al. (2019) using a MARBL-4p1z configuration that includes coccolithophores. Relative to a case with no UV radiation ($E^*_{inh}(0)$), small phytoplankton and diazotrophs experience declines in NPP in the tropical and subtropical regions, where they have higher biomass under
normal conditions relative to other regions; small phytoplankton have increased NPP in the subpolar regions, at the expense of coccolithophore NPP, indicating that decreased fitness of coccolithophores in the regions they are most abundant is opening up habitat for small phytoplankton productivity under UV stress (Figure 4b).



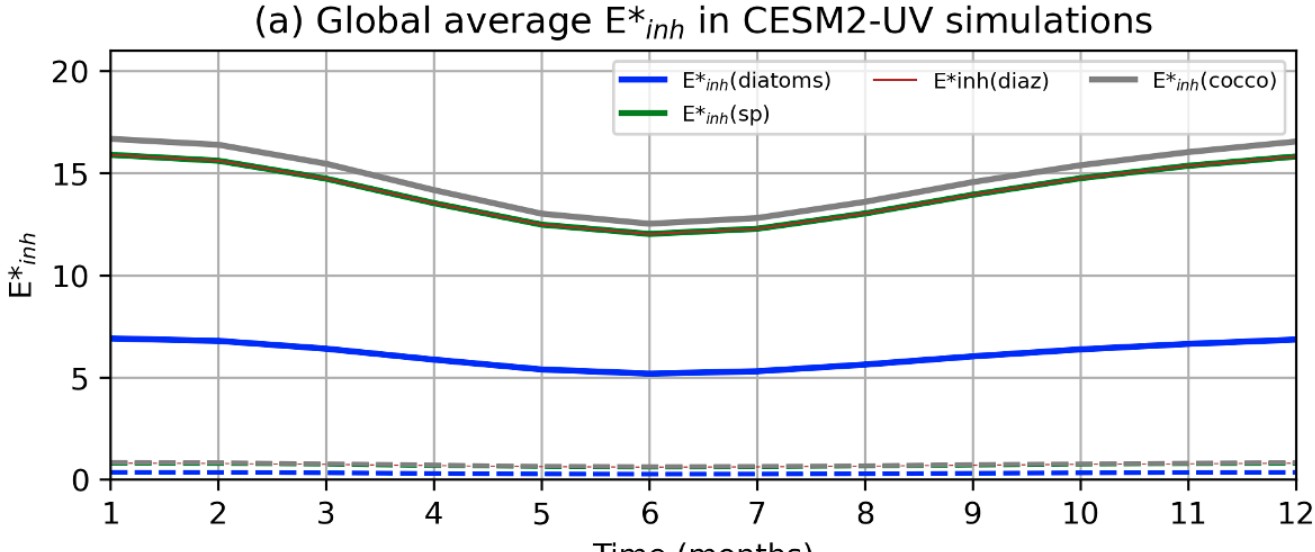

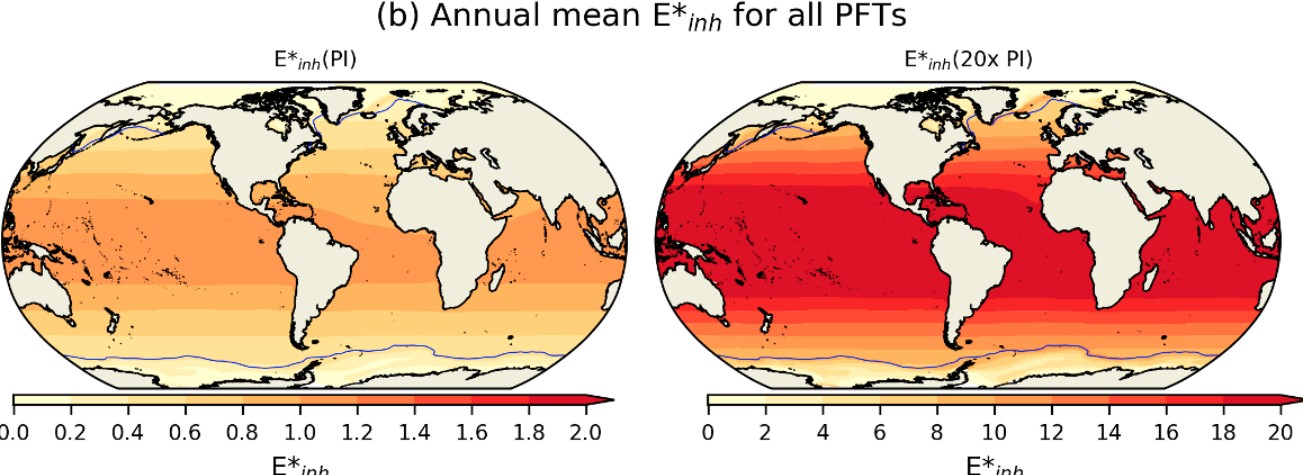

**Figure 3.** (a) Global average $E_{inh}^*$ for each PFT for both $E_{inh}^*$(PI) (dashed) and $E_{inh}^*$(20x PI) (solid). (b) The spatial distribution of total average $E_{inh}^*$ for all PFTs under (left) $E_{inh}^*$(PI) and (right) $E_{inh}^*$(20x PI). Maximum annual sea ice extent is indicated by the blue solid line. Higher values of $E_{inh}^*$ indicate greater plankton limitation.





While $E_{inh}^*$ and its change with UV radiation is approximately zonally homogeneous (Figure 3b), the response of phytoplankton NPP to enhanced UV radiation in the $E_{inh}^*$(20x PI) case is zonally heterogeneous, especially for diatoms and

coccolithophores (Figure 4c). Under elevated UV radiation ($E_{inh}^*$(20x PI)), diatom NPP declines slightly relative to preindustrial NPP in some areas, but increases in many parts of the equatorial Pacific. Here, small phytoplankton experience a large magnitude decrease in NPP with increasing UV radiation (Figure 4c), relieving nutrient limitation and allowing for the growth of diatoms. At 20x PI UV levels, coccolithophores show large declines in NPP relative to the preindustrial UV case in the subpolar Southern Ocean, North Pacific, and North Atlantic. UV inhibition in the top ∼15 m of the ocean drives most of the

global decline of phytoplankton NPP with increasing UV radiation, whereas phytoplankton NPP below this depth tends to increase. Figure 5 shows the UV-radiation driven change in global NPP at each depth level and month over the course of 5 years ($E_{inh}^*$(20x PI) - $E_{inh}^*$(PI); expressed as percent change). The top layer of the ocean (mid-point of 5 m depth) experiences a 35% reduction in NPP, compared to only a 1-2% decrease in NPP in the model's $2^{nd}$ layer (mid-point of 15 m), and a 6.5% increase in NPP in the model's $3^{rd}$ layer (mid-point of 25m) (Figure 5). UV-driven decreases in phytoplankton biomass (not

shown; similar to NPP) in the top two layers allow for an increase in PAR at deeper layers due to a reduction in phytoplankton shading. UV-driven decreases in nutrient uptake in the surface layers (not shown) relieves nutrient limitation throughout the mixed layer.

At high latitudes, NPP below the surface can be enhanced by more than 20% under elevated UV radiation (Figure 5b). While high albedo sea ice offers protection from UV radiation in CESM2-UVphyto, phytoplankton productivity underneath

thin sea ice in the Southern Hemisphere is still reduced by more than 20% from August to October in $E_{inh}^*$(20x PI) compared to $E_{inh}^*$(PI). The reduction of near surface phytoplankton productivity promotes deeper blooms of Southern Ocean phytoplankton productivity from December to March by allowing for PAR to penetrate deeper into the column, corresponding to more than a 10% increase in NPP below 35 m. This signal can be seen in the annual mean total phytoplankton change in NPP in Figure 5b. Subsurface increases in productivity partly compensate for near surface phytoplankton NPP decreases following UV radiation

increases, but phytoplankton NPP declines near the surface dominate the upper ocean integrated net change in phytoplankton productivity in response to UV radiation.

Coccolithophores are uniquely sensitive to UV radiation when $CO_2$ is elevated, which is represented in CESM2-UVphyto with enhanced UV inhibition of coccolithophore growth rates scaled by the inverse of PIC/POC$^{(1/6)}$ (see Section 2.4). Globally integrated coccolithophore NPP, while not a major driver of total phytoplankton NPP in our model (Figure 2b), is nevertheless

influenced by UV radiation. In the $E_{inh}^*$(PI) simulation, global coccolithophore NPP is 3.5 Pg C yr$^{-1}$, decreasing to 2.9 Pg C yr$^{-1}$ in the $E_{inh}^*$(20x PI) simulation, a ∼17% decline (Figures 2b,6). As atmospheric $CO_2$ increases, UV radiation engenders larger decreases in global coccolithophore NPP: a ∼20% decrease at 400 ppm, a ∼24% decrease at 600 ppm, and a ∼25% decrease at 900 ppm in the $E_{inh}^*$(20x PI) simulation relative to the $E_{inh}^*$(PI) simulation (Figure 6). As atmospheric $CO_2$ and thus the aqueous $CO_2$ concentration increases, the PIC/POC distribution in the modeled ocean changes (Figure 7), exposing

a greater proportion of coccolithophores to UV radiation. PIC/POC shows a bimodal distribution, with peak densities at 0 and 1.0 (Figure 7). With increasing $CO_2$, a greater proportion of thin-shelled coccolithophores are characterized as "naked" (PIC/POC<0.05), while the thick-shelled coccolithophores (PIC/POC≈1.0) see a shift in their distribution to lower PIC/POC





**Figure 4.** (a) The spatial distribution of 5 year annual mean productivity vertically integrated over the top 150 m (g C m$^{-2}$ yr$^{-1}$) in the $E*_{inh}$(PI) simulation for small phytoplankton, diatoms, coccolithophores, and diazotrophs. (b) Percent change in annual mean NPP for $E*_{inh}$(PI) - $E*_{inh}$ (0) and (c) $E*_{inh}$(20x PI) - $E*_{inh}$(PI) for small phytoplankton, diatoms, coccolithophores, and diazotrophs.

.

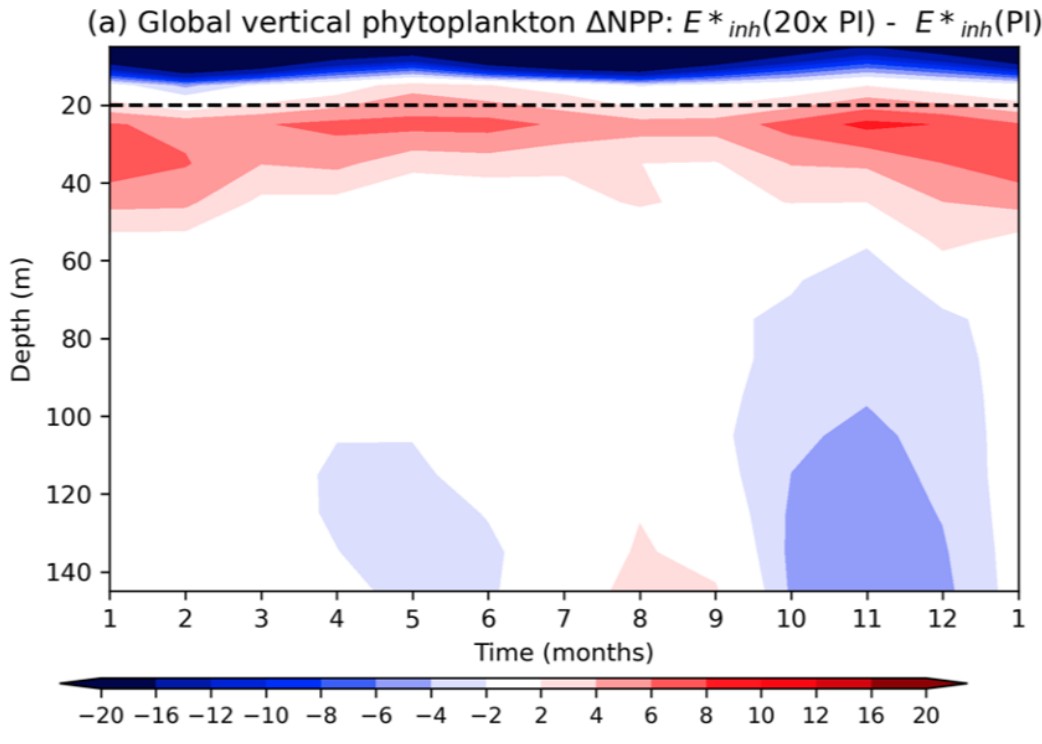

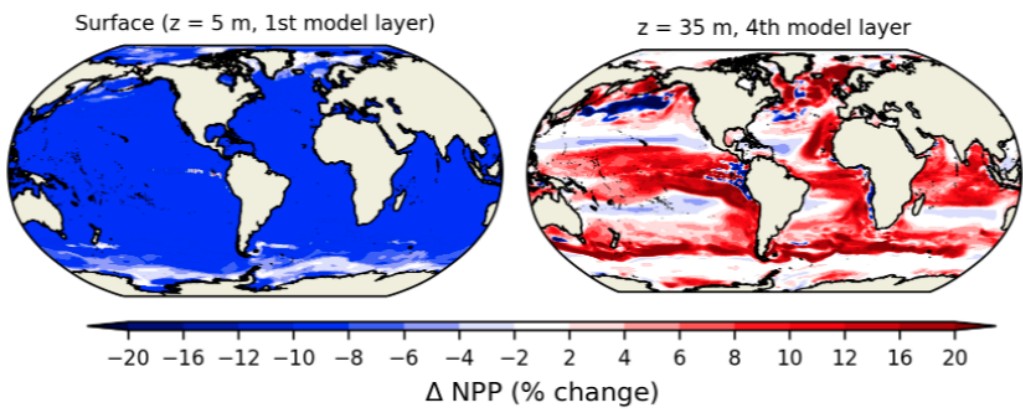

**Figure 5.** (a) UV-driven change in global-mean NPP (g C $yr^{-1}$ $m^{-1}$) as a function of depth, calculated as the NPP from the $E*_{inh}$(20x PI) simulation minus the NPP from the $E*_{inh}$(PI) simulation over 5 years of simulation. The dashed black line represents the average depth where $E_{inh}^*$ attenuates to 1% of its value at the surface. (b) Map of annual mean NPP percent change from the $E*_{inh}$(20x PI) simulation minus the NPP from the $E*_{inh}$(PI) simulation at (left) 5m and (right) 35 m depth over 5 years of simulation.



values, akin to shell thinning (Krumhardt et al., 2019). This shell thinning makes the coccolithophores more susceptible to UV radiation, and thus coccolithophores experience larger UV-driven productivity declines when $CO_2$ is elevated (Figure 6).

In a simulation without PIC/POC scaling forced by $E^*_{inh}$(PI) radiation and $CO_2$=284 ppm, globally integrated total phytoplankton NPP is 55.6 Pg C yr$^{-1}$, less than a 1% change from the simulation with PIC/POC scaling. Despite little change in the sum of all phytoplankton, global coccolithophore NPP declines by approximately 8% (3.8 to 3.5 Pg C yr$^{-1}$, see Figure 6) from $E^*_{inh}$(0) to $E^*_{inh}$(PI) with PIC/POC scaling, compared to 0.3% without. Small phytoplankton and diatoms experience increases in NPP when coccolithophore growth rates are scaled by the inverse of PIC/POC$^{(1/6)}$ even under pre-industrial UV

radiation (not shown). In the PIC/POC scaled $E*_{inh}$(20x PI) simulation, globally integrated total phytoplankton NPP declines to 51.2 Pg C yr$^{-1}$, less than a 0.5% change from a $E*_{inh}$(20x PI) simulation not scaled by coccolithophore PIC/POC. Coccolithophore productivity declines by 25% with PIC/POC scaling in the simulation with $E^*_{inh}$(20x PI) forcing compared to one with $E^*_{inh}$(PI) simulation. Prior to the implementation of PIC/POC scaling, global coccolithophore NPP declined by only 5% under $E^*_{inh}$(20x PI) when compared to $E^*_{inh}$(PI). Coccolithophores exhibit relatively high sensitivity to UV radiation compared

to other PFTs even under pre-industrial $CO_2$ concentrations using scaling derived from laboratory studies (Xu et al., 2016).

Small phytoplankton, diatoms, and diazotrophs all benefit in response to diminished coccolithophore growth, and global integrated total phytoplankton NPP does not vary significantly as $CO_2$ increases. Because coccolithophores with lower PIC/POC values are eliminated by high UV radiation under the scheme with PIC/POC scaling, a positive shift occurs in the globally averaged and biomass weighted coccolithophore PIC/POC over time under $E^*_{inh}$(20x PI). After a year of $E^*_{inh}$(20x PI) forcing,

thin shelled coccolithophores are seemingly eliminated, leaving behind thicker shelled coccolithophores in warmer locations. This shift leaves behind a smaller, but seemingly more resilient population of coccolithophores.

We further assess the spatial heterogeneity in the phytoplankton NPP response to UV radiation under different atmospheric $CO_2$ concentrations using the core biomes defined by Fay et al. (2014). In general, coccolithophores tend to be most negatively impacted by UV radiation in the seasonally ice-covered and subpolar biomes of the north Pacific, north Atlantic, and Southern

Ocean (Figure 8a). The abundance of coccolithophores in these biomes, coupled with their relatively low PIC/POC values mean that coccolithophores are especially responsive to UV radiation increases here. Coccolithophores growing under elevated atmospheric $CO_2$ concentrations (and thus have lower PIC/POC values), experience higher levels of UV inhibition when UV is elevated and see even larger declines in NPP in the supolar north Pacific and Southern Ocean (Figure 8b). For example, in biome 16 (Southern Ocean Subpolar Seasonally Stratified), coccolithophore NPP declines by over 50% in December and

January, which enhances diatom NPP in January and February (Figure 8c). In the tropical and subtropical biomes, in contrast, coccolithophore NPP increases with increasing atmospheric $CO_2$, even as UV radiation increases, driving decreases in small phytoplankton and diatom NPP (Figure 8b).

Figure 8c shows monthly coccolithophore NPP for biome 16 across varying levels of $E^*_{inh}$. The coccolithophore bloom maximum occurs in February to March and is dampened by UV inhibition, especially under PIC/POC scaling, but there is

no change in the timing of the bloom. No change in the timing of blooms is also true for all biomes that experience this bloom behavior, including biomes 1 (North Pacific Ice), 5 (West Pacific Equatorial), 7 (South Pacific Subtropical Permanently Stratified), 8 (North Atlantic Ice), 15 (Southern Ocean Subtropical Seasonally Stratified), and 16 (Southern Ocean Subpolar



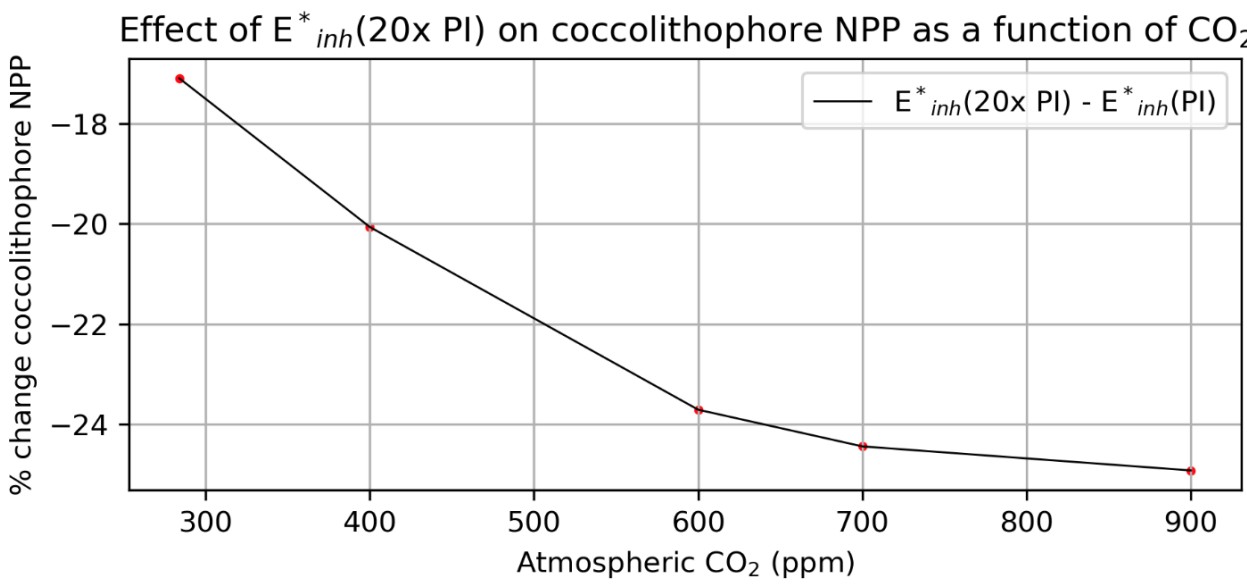

**Figure 6.** Annual mean coccolithophore net primary productivity percent change under simulation with $E*_{inh}$(20x PI) forcing compared to $E*_{inh}$(PI) with varying atmospheric $CO_2$.

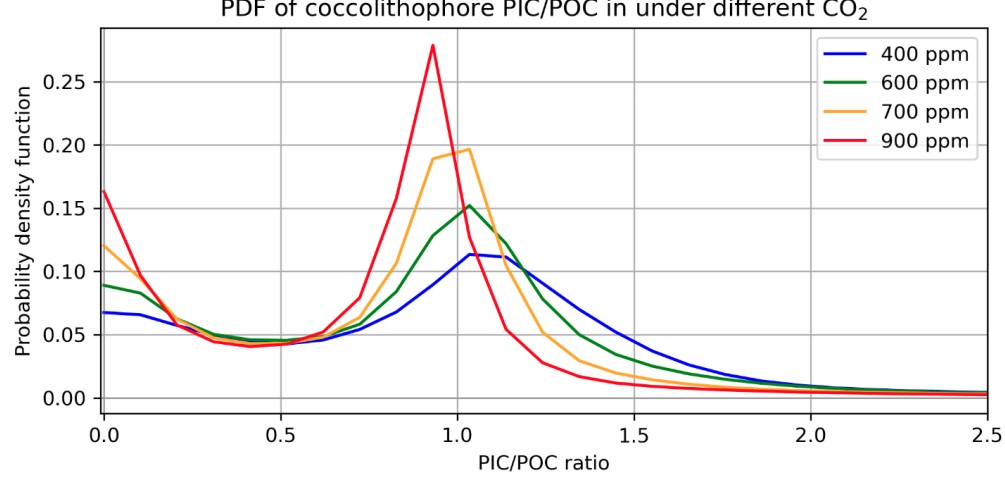

**Figure 7.** The probability density function for annual mean PIC/POC, weighted by coccolithophore biomass, for a 10-year spin-up where $CO_2$ is 400 ppm, 600 ppm, 700 ppm, and 900 ppm.



Seasonally Stratified). Several biomes exhibit a noticeable increase in productivity despite UV inhibition, including biomes 3 (North Pacific Subtropical Seasonally Stratified), 10 (North Atlantic Subtropical Seasonally Stratified), and 12 (Atlantic

Equatorial). Prior to PIC/POC scaling, coccolithiphores in biome 16 experienced a 14% decline in NPP under $E_{inh}^*$(20x PI) compared to $E_{inh}^*$(PI), which is in the accepted range of total NPP decline possible under the effect of the ozone hole, a smaller perturbation compared to the UV pulse in our experiments. Under PIC/POC scaling, coccolithophore NPP declines by 60% in the same biome, demonstrating increased vulnerability in the coldest biomes.

As $CO_2$ is increased to 400 ppm, coccolithophores in biome 16 benefit from relieved carbon limitation more than they are

impacted by thinner coccospheres. When $CO_2$ is increased to 900 ppm, coccosphere thinning dominates and most high latitude regions experience a decline in coccolithophore productivity, while coccolithophores in subtropical and tropical biomes tend to benefit more from relieved carbon limitation (Figure 8b). Overall, the greatest percent change in coccolithophore NPP due to UV radiation occurs at high latitudes and parts of the equatorial Pacific, where cold temperatures and/or upwelling of carbon rich waters result in lower PIC/POC values, thinner coccospheres, and enhanced UV inhibition. In these same regions, small

phytoplankton benefit and experience an increase in NPP, despite relatively high UV radiation.

## 4   Discussion

Changes in phytoplankton NPP are simulated in response to a pulse of UV radiation, with variations that occur as a function of latitude, depth, temperature, nutrient availability, and other phytoplankton characteristics. Globally integrated small phytoplankton NPP is particularly impacted by its large presence at low latitudes, where the simulated pulse of UV radiation is at

its greatest. In the simulations examined here, small phytoplankton begin to be able to compete in the Southern Ocean where the decline of coccolithophores has relieved nutrient limitation in spite of temperature limitation. In an Antarctic ozone hole scenario where UV radiation is enhanced in the Southern Ocean, diatoms and coccolithophores may be more impacted. We do not conduct these simulations here but provide a tool for such experiments in the future.

Few examples of extreme UV pulses have occurred during the existence of modern observation systems for validation

purposes, but our UV pulse simulation results generally fall within the large uncertainty range of existing laboratory and observational studies of phytoplankton under ambient UV radiation (Helbling et al., 1992; Smith et al., 1992), as well as the Southern Hemisphere ozone hole (Prézelin et al., 1994). In simulations with extreme levels of UV radiation at high latitudes, coccolithophore productivity responses to UV radiation are on the high end of existing observational studies which typically involved lower amounts of UV radiation.

The development of CESM2-UVphyto is challenged by our understanding of how phytoplankton respond to increased UV radiation. The calculation of UV damage ($E_{inh}^*$) in CESM2-UVphyto is specific to each phytoplankton type, yet only a small number of laboratory studies report BWFs for the modeled PFTs. No BWF was available for diazotrophs and only one BWF could be found for coccolithophores. Over longer simulations under pre-industrial UV radiation, inaccuracy in the BWFs may lead to drifts in nitrogen, alkalinity, calcium carbonate, etc. compared to the observed ocean. However, pre-industrial levels of

UV radiation are unlikely to cause significant drifts due to the relatively low impacts to phytoplankton productivity. Further

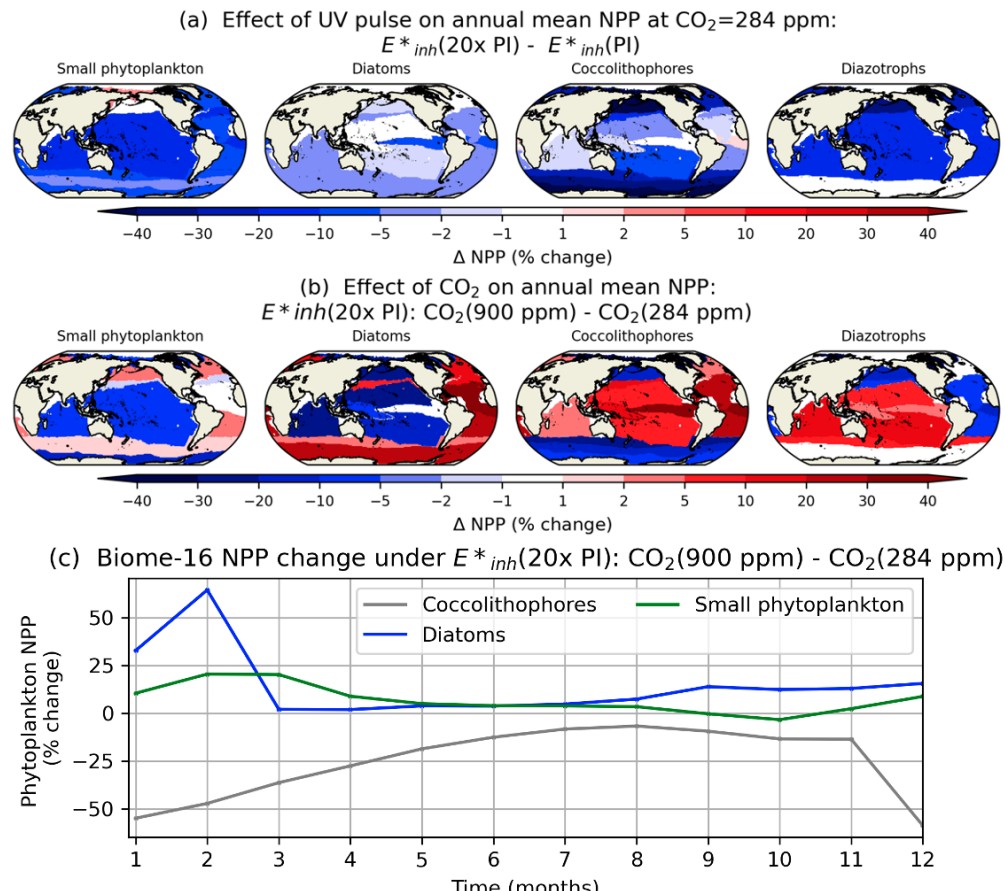

**Figure 8.** (a) Annual mean percent change in NPP across 17 biomes under $E_{inh}^*$(20x PI) - $E_{inh}^*$(PI) and with $CO_2$=284 ppm. (b) Annual percent change in NPP across 17 biomes under $E_{inh}^*$(20x PI): $CO_2$(900 ppm) - $CO_2$(284 ppm). (c) Phytoplankton NPP annual mean percent change for biome 16 (Southern Ocean Ice) under $E_{inh}^*$(20x PI) for $CO_2$ = 900 ppm compared to $CO_2$ = 284 ppm.

research developing BWFs with laboratory studies that are more tailored to the species used to represent each PFT in MARBL would narrow uncertainties in simulating UV inhibition of photosynthesis.

CESM2-UVphyto horizontal and vertical resolution may also play a role in the fidelity of our simulations. At 1 degree nominal horizontal resolution, mesoscale features and some coastal processes are not represented, an absence that potentially
impacts the timing and spatial scale of phytoplankton blooms. Scaling UV radiation penetration into the ocean based on wave properties was not possible in CESM2-UVphyto, but higher resolution modeling with parameterizations accounting for the scattering of UV when encountering waves and whitecaps could improve the simulation of UV radiation penetration into the ocean. We expect regions with significant wave activity to likely experience an overestimate in UV inhibition, while calmer, subtropical waters likely experience an underestimate in our model simulations. Finally, because UV attenuates so quickly
with depth, the available 10 m vertical spacing in CESM2-UVphyto may produce small inaccuracies in UV inhibition of



photosynthesis that can affect the vertical profile of phytoplankton and as a result, the shading of PAR and PAR amounts deeper in the water column. Zooplankton biomass is not considered in shading equations for PAR or UV radiation, potentially leading to small errors in radiation propagation. The absence of chlorophyll in zooplankton minimize its effectiveness at intercepting PAR, but its effects on UV radiation is less clear.

## 5  Conclusions

We have implemented inhibition of photosynthesis from UV radiation for four different types of phytoplankton in the versatile ocean biogeochemistry model MARBL, which can be incorporated into a number of global climate models. The implementation requires the computation of photosynthetic UV damage using biological weighting functions integrated over the wavelengths of UV radiation within the atmospheric model. We explored a large parameter space of global $E_{inh}^*$ values to understand the performance of our modifications at extremes.

CESM2-UVphyto is the first fully coupled Earth system model to calculate and consider UV inhibition of photosynthesis among phytoplankton. Increased UV radiation from the ozone hole likely impacted Southern Ocean phytoplankton, but only simple models have been used to quantify this response. UV radiation may have shaped the recovery of ecosystems during the extinction event at the K-Pg boundary, but simulations of this event have not typically included the role of increased UV radiation after an asteroid impact. CESM2-UVphyto is capable of simulating the emissions from an asteroid impact and the hypothesized pulse of UV radiation afterwards. Simulating the impact of a pulse of UV radiation and its role towards an extinction event in an Earth system model is one example of a use case that can help inform interpretations of proxy records of marine organisms in the years afterwards. Furthermore, quantifying mechanisms of past extinction provides context for ongoing anthropogenic climate change which may involve enhanced UV inhibition in increasingly stratified ocean layers. Other stratospheric aerosol injection events such as volcanic eruptions, large-scale wildfires, geoengineering, or even nuclear war may pose further risks to the stratospheric ozone layer. CESM2-UVphyto provides a modeling tool for quantifying ocean ecosystem impacts of these events.

*Code and data availability.*

The code modifications to the CESM are stored on GitHub at https://github.com/coupewx/CESM2-UV/tree/coupewx-cesm2-uvphyto-01. The data used to produce figures in the main text and supplemental is archived on Zenodo (DOI: 10.5281/zenodo.11127431). A Jupyter notebook to generate all figures can be found at

https://github.com/coupewx/CESM2-UV/blob/master/notebooks/manuscript_figures.ipynb.

*Author contributions.*



NL conceptualized the study. CB added TUV to the atmospheric model. LG, JC, and NL determined biological weighting functions for use in MARBL. JC implemented biological weighting functions into TUV. JC designed and implemented UV inhibition in MARBL with assistance from ML. KK provided experimental 4p2z version of MARBL. JC ran the model simulations. JC analyzed simulation results and prepared the manuscript. JC, et al., assisted in preparing and reviewing the manuscript.

*Competing interests.*

We declare no competing interests.

*Acknowledgements.* Model simulations were performed on the NSF National Center for Atmospheric Research Cheyenne and Derecho supercomputers. This material is based upon work supported by the National Center for Atmospheric Research, which is a major facility sponsored by the National Science Foundation under Cooperative Agreement No. 1852977. Funding for work was provided by the National Science Foundation FRES #2021686. Funding for work was also provided by the Future of Life Institute.





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
