# Peer review of "Sunburned plankton: Ultraviolet radiation inhibition of phytoplankton photosynthesis in the Community Earth System Model version 2"

_Geoscientific Model Development, 2024_

## Referee Comment (RC1)

Comments on GMD-2024-94: *Sunburned plankton: Ultraviolet radiation inhibition of phytoplankton photosynthesis in the Community Earth System Model version 2*
Authors:  Coupe et al.

General Comments – The authors have implemented for the first time an earth system model that includes the effect of solar UV irradiance on the photosynthesis of marine phytoplankton.  Although there have been several modeling exercises that address the effect of UV inhibition of photosynthesis on, e.g., daily areal productivity for global or regional (e.g. Southern Ocean) basis, this is the first time those responses have been integrated into a full ecosystem model with the provision of feedback effects and shifts in taxonomic composition.  This new model is potentially useful in inferring what effects UV currently has on the marine pelagic ecosystem as well as how these effects may change in response to various extreme events or climate manipulations.  An additional use case would be to compare the model output for scenarios with and without the controls on ozone depleting substances imposed by the Montreal Protocol.  Such global assessments of the "world avoided" have thus far only been conducted for carbon cycling in terrestrial ecosystems (Young et al. 2021).

While I applaud the work of the authors in structuring the model, the implementation of the biological weighting functions and penetration of weighted UV radiation has several deficiencies.  I expect that the model can be corrected to address these problems (detail provided below), so that the modeling and assessment community can have a CESM2-UVphyto that is consistent with our current understanding of UV effects on phytoplankton.

Specific Comments

1)  A general point that should be made clear to any user of the model is that sensitivity to inhibition by UV irradiance is a physiological characteristic that is as variable as any other parameter of phytoplankton photosynthesis.  Sensitivity is variable mainly because net inhibition reflects that balance between damage and repair processes (e.g. Neale and Kieber 2000).  Variability on the damage side primarily derives from physical characteristics – e.g. the optical characteristics and cell dimension – these most often vary in a narrow range for any one taxa.  However, repair processes can vary considerable depending on growth conditions.  In this version of CESM2-UVphyto, Biological Weighting Functions (BWFs) are fixed irrespective of growth conditions (except for $CO_2$ – see below).  This is an inconsistency in the code since the MARBL model does incorporate photoadaptation of photosynthesis in general through the C:Chl ratio, responding to growth irradiance, temperature and nutrients.   It shouldn't be surprising that sensitivity to UV inhibition is also affected by these factors and there are many studies that confirm this beyond the studies cited for the model (see bibliography below). Where variation is known, e.g. for temperature and growth irradiance (cf. Neale and Thomas 2016), the code should account for it.  If  UV inhibition response is known only for one growth condition, the model should warn of increased uncertainty for predictions beyond the experimental conditions used for determination.

2) The one condition for which the model varies sensitivity to UV is in relation to atmospheric $CO_2$, which changes the $pCO_2$ and pH of ocean water. Ironically, the BWFs for the microalga chosen to simulate the effect of changing $CO_2$ show no evidence that they are affected by elevated $CO_2$. Lorenzo et al. (2019) compared the BWFs of *E. huxleyi* grown at equilibrium with atmospheric $CO_2$ of 400 and 800 ppm and found no difference even though there were changes in the coccoliths. Xu et al (2016) observed somewhat contrasting results in that a calcified strain was more resistant to UV than a naked strain. They concluded that coccoliths have an effect protecting against UV, however the experiment was not controlled in the sense that there were several differences between the + and – UV treatments, besides UV (including strain, PAR level, variability of exposure). These differences don't exclude that coccoliths perform a screening function, however the study of Lorenzo et al was performed under controlled conditions so that $CO_2$ was the only factor that varied. In this case, the changes in coccoliths were insufficient to affect sensitivity to UV or if the change in coccoliths did allow more damaging UV to reach the cell, the effects were compensated by enhanced repair capabilities (this is discussed by Lorenzo et al). In summary, it would be inappropriate to vary the Einh (* omitted for convenience) computed from the Lorenzo et al BWF according to the PIC/POC ratio when Lorenzo et al did not observe an effect. As an aside, other taxa do show increased sensitivity to UV under $CO_2$ enhancement and it would be interesting to evaluate their responses in the context of the model (see Sobrino et al 2008).

In principle, I expect that the model code can be changed to accommodate this variability, which I encourage the authors to do. But I also recognize that the BWFs used in the case studies shown could be regarded as a "proof of concept" choices. This is defensible as along as it is made clear that results could be quite different for other choices, even for taxa within the same PFT group. However, independent of the choice of BWF, there are several steps described in the calculation of inhibition of photosynthesis section that are incorrect and lead to results that are inconsistent with current understanding of UV effects.

3) The propagation of Einh through the water column cannot be approximated with the attenuation of a single wavelength (Eqs 2 and 3). Although it is often used as a proxy for the attenuation of DNA damaging UV-B, the attenuation coefficient at 305 nm is inappropriate for propagating Einh(z) because most of the weight derives from UV-A. As a result, Einhz propagated with Kd305, declines with depth much faster than that calculated with a fully spectral resolved $Kd(\lambda)$, as shown in this example for clear oceanic water :

[Figure]

Figure 1: Depth profiles of Einh estimated using either a single attenuation coefficient (Kd(305)) applied to weighted irradiance at the surface (solid line) or using spectral attenuation coefficients (dashed line). Spectral irradiance and attenuation coefficients in the Pacific at 15°S at midday were estimated as described by Neale and Thomas (2016) and Einh calculated using a BWF for *Synechococcus* (ML@26°C, Neale et al. 2014). For this profile, Kd (305) was 0.139 m$^{-1}$.

The depth to which UV inhibition affects photosynthesis in clear ocean waters is much deeper than 16 m (see also Fig. 2 in Neale and Thomas 2016). The effective Kd(z) (=-ln(Einh(z+1)/Einh(z)) in this example is similar to that of Kd($\lambda$=327nm) at the top of profile but changes (decreases) progressively to be similar to Kd($\lambda$=388 nm) at 100 m, with about 2.4x change in apparent Kd over the profile.

4) Therefore, a more wavelength resolved approach is needed to propagate Einh. The values of Overmans and Agusti (2020) for coral reef areas in the Red Sea are inappropriate to apply over the whole ocean (Eq 2). Many areas of the ocean have more UV transparency than the Red Sea. Tedetti and Sempéré (2007, Table 2) reviewed global measurements of UV penetration and report that most open ocean waters have, e.g., 10%UV-B depths > 8 m. The maximum 10%UV-B depth possible from the Overmans and Agusti equation is 2.3/.29 =7.9 m. UV penetration is higher in the open ocean because it is further from land and has lower concentrations of colored dissolved organic matter (CDOM) than the Red Sea. CDOM is more important in determining UV transparency than Chl. One possible approach to a more representative spectral Kd is to use the equation of Lee et al (2013) (see their Eq. 5), estimating the IOPs of absorption and backscattering based on MARBL parameters – mainly Chl, POC and DOC.

5) BWFs in the literature are part of specific integrated photosynthesis-irradiance relationships (BWF/P-I models), and only correctly estimate UV inhibition if implemented using the relationship for which they were defined. There are three BWF/P-I models : E, T and Emax (see comparative discussion in Lorenzo et al. (2019), and each BWF used in the CESM2-UVphyto case studies uses a different linked model. Particularly important is the difference between the E model used for the Diatom (*Phaeodactylum*) BWF (1/[1+Einh] dependence) and the T model used for the Coccolithophore (*E. huxleyi*) BWF (1/Einh

dependence).  Evaluating UV inhibition by substituting the *E. huxleyi*  Einh into the E model predicts more inhibition than if implemented with the correct T model (Fig. 2)

[Figure]

Figure 2.  Examples of the depth profiles of $\gamma_{UV}$ evaluated with three different BWF models using spectral irradiance in the Pacific at 15°S at midday (cf. Fig 1).  (1) Syn-Uses a BWF for *Synechococcus* (ML@26°C, Neale et al. 2014) using the same Einh in either the Emax (solid green) or E model (dashed green); (2) Phaeo- *Phaeodactylum sp.* using the E model (brown line) and (3) Ehux - *Emiliana huxleyi* using the T model (black solid) or same Einh in the E model (dashed black line).

For the T model prediction of *E. huxley*i inhibition (Fig. 2), there is a sharp decline in inhibition with depth as this model was optimized to best predict responses at high exposure.  The same Einh evaluated with the E model has lower $\gamma_{UV}$ (more inhibition) and penetrates deeper.  Lorenzo et al acknowledged that there probably is some inhibition at lower exposures (and deeper in the water column), but this could not be resolved given the variability of the laboratory measurements, so below a certain depth (23 m in the above) there is no effect.  In the example, the overall inhibition of midday productivity over the upper 100 m, given the P-I parameters in Lorenzo et al, is only 11% with the T model, but increases to 30% if the T model Einh is used to predict $\gamma_{UV}$ using Equation 4 (E model).  This directly impacts conclusions drawn from the reported case studies that coccolithophores are the most sensitive to inhibition by UVR.  Higher Einh, per se, doesn't mean more inhibition if the models are different.

6) On the other hand, for the small prokaryotes, *Prochlorococcus* and *Synechococcus*, response at low exposure (which varies as 1/[1+Einh]) could be resolved and distinguished from that at high exposure (1/Einh dependence), the two responses are combined in the Emax model (Neale et al 2014).  The BWF/PI for both Pro and Syn use the Emax model, which in this case leads to more inhibition for the Einh(PI) case compared to using the 1/[1+Einh] form over the whole water column (Fig. 2).  Use of the correct model combined with more accurate Einh propagation should result in much higher relative effect of UV on ocean productivity for Einh(PI) than the ~1% effect in the reported case studies. The cited range of 7-28% inhibition reported by Neale and Thomas 2016 is the inhibition of integrated midday production by UV (vs only PAR) in their simulations of the Pacific for current conditions, not what was observed in the laboratory. Similarly, estimates of around 7%

inhibition of daily integrated production by UVR have been obtained by other approaches, e.g. Cullen et al. (2012) and Moreau et al. (2015).

7) The BWF function chosen for small phytoplankton is for *Prochlorococcus*, which is much more sensitive to inhibition by UV radiation than other picophytoplankton (Neale and Thomas 2016). It is better consider Pro a separate case. The BWFs for *Synechococcus* is probably more representative of picophytoplankton overall as well as diazotrophs.

8) Equation 1 should include a term for inhibition by PAR ($\varepsilon_{PAR}*E_{PAR}$ cf. Eq. 3 in Neale and Thomas 2016). PAR inhibition is significant for the cases of *E. huxleyi*, *Prochlorococcus* and *Synechococcus*, and other published BWFs. No PAR inhibition was defined for *Phaeodactylum*, that is partly related to the lower $E_{PAR}$ exposures used in this first experimental determination of the BWF.

9) Elevated UV radiation. Because most inhibition is caused by UV-A radiation, Einh at the surface for known BWFs will never be increased by a factor of 20 vs Einh(PI), even if 95% of stratospheric ozone is destroyed (total ozone column would be reduced by a lower fraction as the increased penetration of UV-C will form ozone at lower altitudes). Thomas et al (2015) treated the case of a gamma-ray burst resulting, briefly (months), in a strong depletion of column ozone by 70% in the region of the strike. At most, this increased the Einh(0) by a factor of 2.63, based on the Phaeo BWF which has probably the largest ratio of sensitivity to UV-B vs UV-A. The multiplier will be less for almost all other phytoplankton especially when PAR inhibition is included. The effect of a gamma-ray burst on Integrated production at midday was also evaluated for *Synechococcus* and *Prochlorococcus* (Neale and Thomas 2016b) and ozone depletion of 70% resulted in at most 3% additional inhibition (vs normal ozone) to the 1% depth of PAR and 7% additional inhibition of productivity integrated over the mixed layer.

10) Yet another approach is needed for polar phytoplankton, for which low light and temperature can result in very low repair rates such that in some cases (mainly deeply mixed zones), inhibition is time dependent (see Neale et al 1998, Smyth et al 2012). Probably polar oceans, especially S. Ocean inside ice-limit, should be treated as a special case.

11) Finally, the CESM2-UVphyto model assumes that the only effect of UV on phytoplankton is through inhibition of photosynthesis. However, UV (more specifically UV-B) also directly damages DNA, inhibiting growth. Too little is known to quantify the importance of this mode of UV effect on a global basis, still it should be recognized that it could be important (see Andreasson and Wangberg 2007).

Summary – The model needs extensive revision after which the case studies can be re-run. At that point the results and conclusions can be re-examined and re-written as needed. Detailed reviews of those sections will be provided then.

Minor and Technical Comments

Line 95 Although stratification has intensified, surface mixed layer depths are not increasing, contrary to the expectations of Gao et al. (2019). See discussion in Neale et al. (2023)

Figure 1 The text in several places states limits to UV-B as 280-315 nm, but this plot has the division between the bands at 320 nm

Line 230 Supplemental Table 1 only lists weights up to 327 nm

Line 230 Relative to the originally published BWFs, weights have been both interpolated and extrapolated.

Line 265 As mentioned above, Lorenzo et al. 2019 did not find an effect of PIC/POC ratio on the BWF. In any case, the treatment effect found by Xu et al (which could be UV as well as other factors) is more accurately defined as 3.5/2=1.75, since the calcified strain grew twice as fast as the naked strain under PAR-only incubator conditions.

Line 520 Arrigo 1994 – journal name missing

Line 620 Neale and Thomas GCB, publication year is 2016

Respectfully submitted,

Patrick Neale
Edgewater, MD USA

References:

Andreasson, K. I. M., & Wängberg, S.-A. (2007). Reduction in growth rate in *Phaeodactylum tricornutum* (Bacillariophyceae) and *Dunaliella tertiolecta* (Chlorophyceae) induced by UV-B radiation. Journal of Photochemistry and Photobiology B: Biology, 86(3), 227-233.

Banaszak, A. T., & Neale, P. J. (2001). UV Sensitivity of photosynthesis in phytoplankton from an estuarine environment. Limnol Oceanogr, 46, 592-600.

Cullen, J. J., R. F. Davis, and Y. Huot (2012), Spectral model of depth-integrated water column photosynthesis and its inhibition by ultraviolet radiation, Global Biogeochem. Cycles, 26, GB1011, doi:10.1029/2010GB003914.

Lee, Z., C. Hu, S. Shang, K. Du, M. Lewis, R. Arnone, and R. Brewin (2013), Penetration of UV-visible solar radiation in the global oceans: Insights from ocean color remote sensing, J. Geophys. Res. Oceans, 118, doi:10.1002/jgrc.20308.

Litchman, E., & Neale, P. J. (2005). UV Effects on photosynthesis, growth and acclimation of an estuarine diatom and cryptomonad. Mar Ecol Prog Ser, 300, 53–62.

Litchman, E., Neale, P. J., & Banaszak, A. T. (2002). Increased sensitivity to ultraviolet radiation in nitrogen-limited dinoflagellates: photoprotection and repair. Limnol Oceanogr, 47, 86-94.

Moreau, S., Mostajir, B., Bélanger, S., Schloss, I. R., Vancoppenolle, M., Demers, S., & Ferreyra, G. A. (2015). Climate change enhances primary production in the western Antarctic Peninsula. Global Change Biology, 21(6), 2191-2205. https://doi.org/10.1111/gcb.12878

Neale, P. J., Banaszak, A. T., & Jarriel, C. R. (1998). Ultraviolet sunscreens in dinoflagellates: Mycosporine-like amino acids protect against inhibition of photosynthesis. J Phycology, 34, 928-938.

Neale, P. J., Cullen, J. J., & Davis, R. F. (1998). Inhibition of marine photosynthesis by ultraviolet radiation: Variable sensitivity of phytoplankton in the Weddell-Scotia Sea during the austral spring. Limnol Oceanogr, 43, 433-448.

Neale, P. J., & Kieber, D. J. (2000). Assessing biological and chemical effects of UV in the marine environment: Spectral weighting functions. In R. E. Hester, & R. M. Harrison (Eds.), Causes and Environmental Implications of Increased UV-B Radiation, 14, (pp. 61-83). Royal Society of Chemistry

Neale, P. J., Pritchard, A. L., & Ihnacik, R. (2014). UV effects on the primary productivity of picophytoplankton: biological weighting functions and exposure response curves of Synechococcus. Biogeosciences, 11, 2883-2895. https://doi.org/10.5194/bgd-10-19449-2013

Neale, P. J., & Thomas, B. C. (2016b). Solar irradiance changes and phytoplankton productivity in Earth's ocean following astrophysical ionizing radiation events. Astrobiology, 16(4), 245-258. https://doi.org/10.1089/ast.2015.1360

Neale, P. J., Williamson, C. E., Banaszak, A. T., Häder, D. P., Hylander, S., Ossola, R., Rose, K. C., Wängberg, S. Å., & Zepp, R. (2023). The response of aquatic ecosystems to the interactive effects of stratospheric ozone depletion, UV radiation, and climate change. Photochemical & Photobiological Sciences, 22(5), 1093-1127. https://doi.org/10.1007/s43630-023-00370-z

Smyth, R. L., Sobrino, C., Phillips-Kress, J., Kim, H.-C., & Neale, P. J. (2012). Phytoplankton photosynthetic response to solar ultraviolet irradiance in the Ross Sea Polynya: Development and evaluation of a time-dependent model with limited repair. Limnol Oceanogr, 57(6), 1602–1618.

Sobrino, C., & Neale, P. J. (2007). Short-term and long-term effects of temperature on phytoplankton photosynthesis under UVR exposures. J Phycol, 43, 426-436.

Sobrino, C., Neale, P. J., & Lubian, L. (2005). Interaction of UV-Radiation and Inorganic Carbon Supply in the Inhibition of Photosynthesis: Spectral and Temporal Responses of Two Marine Picoplankters. Photochem Photobiol, 81, 384–393.

Sobrino, C., Ward, M. L., & Neale, P. J. (2008). Acclimation to elevated carbon dioxide and ultraviolet radiation in the diatom Thalassiosira pseudonana: Effects on growth, photosynthesis, and spectral sensitivity of photoinhibition. Limnol Oceanogr, 53, 494-505.

Tedetti, M., & Sempéré, R. (2006). Penetration of Ultraviolet Radiation in the Marine Environment. A Review. Photochemistry and Photobiology, 82(2), 389-397. https://doi.org/http://dx.doi.org/10.1562/2005-11-09-IR-733

Young, Paul J.; Harper, Anna B.; Huntingford, Chris; Paul, Nigel D.;Morgenstern, Olaf; Newman, Paul A.; Oman, Luke D.; Madronich, Sasha;Garcia, Rolando R.. 2021. The Montreal Protocol protects the terrestrial carbon sink. Nature, 596 (7872). 384-388. https://doi.org/10.1038/s41586-021-03737-3

---

## Referee Comment (RC2)

Sunburned plankton: Ultraviolet radiation inhibition of phytoplankton photosynthesis in the Community Earth System Model version 2, by Joshua Coupe et al.

This paper presents modifications made to CESM2 to enable the simulation of UV-B (280-315 nm) and UV-A (315-400 nm) propagation through the ocean and UV inhibition of phytoplankton photosynthesis. Biological weighting functions (BWF) that determine UV inhibition were incorporated into the Tropospheric Ultraviolet and Visible (TUV) model, in order to calculate spectrally-integrated BWFs at an hourly frequency. The weighting functions of inhibition were then applied in the ocean biogeochemistry model MARBL, which represents 4 phytoplankton PFTs – small phytoplankton, diatoms, diazotrophs, and coccolithophores; and 2 zooplankton PFTs. In MARBL, coccolithophore shell thickness can be impacted by $CO_2$ levels.

The damage functions were based on the few available studies of productivity responses to changes in UV radiation. There are several pertinent applications and needs for Earth system models to represent these impacts, and this study is an important step forward for the CESM community and would be of interest to other modelling groups. The paper describes the modelling approach clearly and the results are described with an appropriate level of detail. My main comments relate to some inconsistencies in the description of the study and results, organization, and framing of the results.

Major comments:
1. Scale of UV changes:
a. How do the PI UV radiation levels compare to present-day? Since occasionally the results are compared to existing studies, it would be helpful to add a discussion of the differences between present-day and preindustrial UV, and how these might impact the results.

b. 20xPI sensitivity study: This is a huge impact (95% global reduction in stratospheric ozone), if Pinatubo reduced NH ozone by 10% and Hunga Tonga reduced tropical ozone by 5% (also see Fleming et al. 2024 for more recent estimate of Hunga Tonga impacts on ozone from the water vapor injected into the stratosphere). I suggest returning to these discrepancies in the discussion, to highlight that the changes discussed would not be expected on a global scale (perhaps unless a very extreme case like asteroid impact or nuclear war), but instead they display the model sensitivities to changes in UV radiation.
c. The discussion in the supplement of ozone hole stress would fit well into the main discussion of the manuscript and further helps put these results into context of historical impacts.

2. Further discussion of the attenuation of radiation with depth: Equations 2-3 and the values of K imply that most UV radiation attenuates by 16 m with low chlorophyll concentrations. But Figure 5 shows changes down to the 30-40m layer (Layer 4). Does the attenuation of PAR with depth follow a different trajectory? It seems to me that if <1% of surface PAR reached below 16m, there would be very minor impacts in the 3$^{rd}$ and 4$^{th}$ layers. What are the changes to PAR at depth in these experiments? Has there been any validation of depth profiles of PAR vs biomass at these depths? (Minor point: Line 374: This states there is an increase in NPP in the 3$^{rd}$ layer (25m), but Figure 5 shows the 4$^{th}$ layer (35m) – it would be better if the text and figures referred to the same layers)

3. As this is a first step in modelling these impacts, I think the authors could draw even more attention in the discussion to the additional modelling and observational studies that would help inform future developments or address any of the limitations in this study. Consider devoting a short sub-section or paragraph in the Discussion to future research priorities to help progress this work.

4. Description of methods: The abstract states that you conducted simulations to calibrate estimates of sensitivity of phytoplankton productivity to UV radiation. But this is not reflected in the paper, since the experiments discussed only used one value of the biological weighting functions. It would be more accurate to say you did these experiments to 'understand' the sensitivity, unless some calibration was done that is not discussed in the paper (in which case, that should be explained in the manuscript). Also Line 485: I do not see evidence of a large parameter space of E*_inh being investigated in this study.

Minor points
Organization: The results seem out of order to me. Figure 4 is helpful for understanding where the different PFTs live, and the absolute impacts of UV radiation on their NPP. This helps with the understanding of the globally averaged results shown in Figure 2. I suggest rearranging and adding subsections, ie: (1) regional impacts of UV radiation on phytoplankton NPP; (2) globally integrated impacts of UV radiation; (3) vertical distribution of impacts; (4) effect of enhanced CO2.

Lines 151-154: What is the motivation for the sensitivity studies with different configurations of MARBL? They are not discussed in the manuscript (with exception of a very small mention of the 3p1z experiment in the results). So either remove the mention of these additional experiments, or update manuscript to include these in your results/discussion and more fully explain how they contributed to the understanding of phytoplankton sensitivities to UV radiation. (Moving the supplement into the main text would help, as this is relatively short and very relevant to the validation of your results)

Equation 1: Over what interval of lambda are these calculated? (What is Δlambda?)

Line 333: Increase in productivity – is this in relative or absolute terms? Only relative terms are shown in Figure 2.

Figure 3: How is the total E_inh calculated, is it weighted by PFT distribution?

Lines 355-360: Here, and elsewhere in the results, I would suggest you keep the description of the patterns and other direct results from these experiments, but leave the comparison to other studies for the Discussion.

Lines 453ish: Mention here the amount of increased UV radiation in this region in these experiments compared to ozone hole or other historical events.

Lines 458-459: Provide a reference here and clarify – this is not clear – the responses are on the high end of observed values of productivity or observed values of productivity changes?

Line 465ish: In addition, I would think that phytoplankton are adapted to ambient UV levels (this is the case for terrestrial plants), which would have an impact on the spatial variation of the inhibition factor / BWFs. Would it be important to consider the ambient levels of UV to which phytoplankton are accustomed, before considering impacts of changes to those levels?

Lines 425-427: are you sure it's this way around? Could it be that the decreases in small phyto and diatoms are making it easier for coccolithophores to live here?

Line 436-437 – provide a reference here, but also consider moving comparisons of results to other studies to the discussion.

Paragraph from Lines 428-438: When discussing PIC/POC scaling, is this for the case of 900 ppm? This is implied since it references Figure 8c. Please clarify.

References

Fleming, E. L., Newman, P. A., Liang, Q., & Oman, L. D. (2024). Stratospheric temperature and ozone impacts of the Hunga Tonga-Hunga Ha'apai water vapor injection. Journal of Geophysical Research: Atmospheres, 129, e2023JD039298. https://doi.org/10.1029/2023JD039298

---

## Community Comment (CC1)

It is unclear to me what assumptions the model makes about the rates of mixing within the mixed layer.

Estimates of photosynthetic rates based on in situ incubations typically involve suspending incubation bottles at fixed depths so that there is no possibility for vertical movement. In 1978 John Marra (Marine Biology 46:203–208) explored the implications of this compared to systematically alternating the irradiance to simulate vertical movement. He found that vertical movement increased production versus keeping the phytoplankton at a fixed depth. I would assume that similar issues would apply to UV light effects. Because the effects of UV are nonlinear, assuming that the phytoplankton move up and down rapidly would likely lead to different conclusions than assuming that they stay at a fixed depth within the mixed layer. It is unclear to me what the model assumes about vertical movement. That might be something to explore in the future.

How to deal with the CFC problem seems to be a work in progress. The initial fix was hydrochlorofluorocarbons (HCFCs), and then came hydrofluorocarbons (HFCs). The former are unsatisfactory because they still contain chlorine, and the latter are unsatisfactory because they are potent greenhouse gases. Both have been mandated to be phased out. In the meantime, the residence time of chlorine in the stratosphere is 40–100 years, which explains why there has been little perceptible improvement in the ozone hole (https://ozonewatch.gsfc.nasa.gov/).

There is a lot of discussion in the paper about coccolithophores. I have attached a very recent paper by Bradley and Laws (Water 16(22): 3184 https://doi.org/10.3390/w16223184) that concerns effects of elevated $CO_2$, temperature, and nutrient limitation on *Emiliania huxleyi*. This is one strain of *E. huxleyi*, and not all strains can be expected to behave in the same way. The PIC/POC ratio was greater than 1.0, and calcification was surprisingly insensitive to increases of $pCO_2$.

---

## Author Comment (AC3)

Author responses to the comments are in blue.

We thank the reviewer for their comments and suggestions.

It is unclear to me what assumptions the model makes about the rates of mixing within the  mixed layer.

Estimates of photosynthetic rates based on in situ incubations typically involve suspending  incubation bottles at fixed depths so that there is no possibility for vertical movement. In  1978 John Marra (Marine Biology 46:203–208) explored the implications of this compared  to systematically alternating the irradiance to simulate vertical movement. He found that  vertical movement increased production versus keeping the phytoplankton at a fixed  depth. I would assume that similar issues would apply to UV light effects. Because the  effects of UV are nonlinear, assuming that the phytoplankton move up and down rapidly  would likely lead to different conclusions than assuming that they stay at a fixed depth  within the mixed layer. It is unclear to me what the model assumes about vertical  movement. That might be something to explore in the future.

POP uses the K-Profile-Parameterization (KPP; Large et al., 1994) to parameterize vertical mixing in the mixed layer. Thus, there are wind driven vertical fluxes. Phytoplankton and zooplankton are tracers that will follow the currents horizontally and vertically in addition to being altered by photosynthesis and grazing, respectively (Long et al., 2021). MARBL has a vertical resolution of ~10 m in the mixed layer and the top two layer midpoints are 5 m and 15 m depth (Long et al., 2021). Sub-grid scale vertical fluxes that may move phytoplankton closer to the surface may not be well represented because of the vertical resolution of the model. This could be solved by simply increasing the vertical resolution of the model. This is beyond the scope of this work, but the methodologies described in this manuscript can easily be applied to a higher

resolution model in the future. The Discussion section includes text about some of these caveats.

"Finally, because UV attenuates so quickly with depth, the available 10 m vertical spacing in CESM2-UVphyto may produce small inaccuracies in UV inhibition of photosynthesis that can affect the vertical profile of phytoplankton and as a result, the shading of PAR and PAR amounts deeper in the water column. Implementation of UV inhibition in a model with higher vertical resolution would likely resolve these processes with greater accuracy."

How to deal with the CFC problem seems to be a work in progress. The initial fix was hydrochlorofluorocarbons (HCFCs), and then came hydrofluorocarbons (HFCs). The former are unsatisfactory because they still contain chlorine, and the latter are unsatisfactory because they are potent greenhouse gases. Both have been mandated to be phased out. In the meantime, the residence time of chlorine in the stratosphere is 40–100 years, which explains why there has been little perceptible improvement in the ozone hole (https://ozonewatch.gsfc.nasa.gov/).

Thank you for your comment. In the text, we have lightly edited the text to clarify that the ozone hole stopped growing in response to the Montreal Protocol, but has not recovered yet.

There is a lot of discussion in the paper about coccolithophores. I have attached a very recent paper by Bradley and Laws (Water 16(22): 3184 https://doi.org/10.3390/w16223184) that concerns effects of elevated $CO_2$, temperature, and nutrient limitation on *Emiliania huxleyi*. This is one strain of *E. huxleyi*, and not all strains can be

expected to behave in the same way. The PIC/POC ratio was greater than 1.0, and calcification was surprisingly insensitive to increases of $pCO_2$.

Thank you for this reference. In MARBL, the existing coccolithophore functional type represents species including Calcidiscus leptoporus, Gephyrocapsa oceania, as well as four different morphotypes of Emiliania huxleyi (Krumhardt et al., 2017; 2019). Together, these cover most parts of the global ocean. The reasoning for considering many species was to capture broad scale responses. Text has been added to the text to make it clear how this was designed and may not consider individual strains.

"In total, the dynamics four morphotypes of *Emiliana huxleyi*, as well as other species such as *Calcidiscus leptoporus* and *Gephyrocapsa oceania* were used to construct the PFT, allowing the model to simulate the general response of coccolithophores to environmental changes (Krumhardt et al., 2019)."

References

Krumhardt, K. M., Lovenduski, N. S., Iglesias-Rodriguez, M. D., & Kleypas, J. A. (2017). Coccolithophore growth and calcification in a changing ocean. Progress in Oceanography, 159, 276–295. https://doi.org/10.1016/j.pocean.2017.10.007

Krumhardt, K. M., Lovenduski, N. S., Long, M. C., Levy, M., Lindsay, K., Moore, J. K., & Nissen, C. (2019). Coccolithophore Growth and Calcification in an Acidified Ocean: Insights From Community Earth System Model Simulations. Journal of Advances in Modeling Earth Systems, 11(5), 1418–1437. https://doi.org/10.1029/2018MS001483

Large, W. G., McWilliams, J. C., & Doney, S. C. (1994). Oceanic vertical mixing: A

review and a model with a nonlocal boundary layer parameterization. Reviews of Geophysics, 32(4), 363–403. https://doi.org/10.1029/94RG01872

Long, M. C., Moore, J. K., Lindsay, K., Levy, M., Doney, S. C., Luo, J. Y., Krumhardt, K. M., Letscher, R. T., Grover, M., & Sylvester, Z. T. (2021). Simulations With the Marine Biogeochemistry Library (MARBL). Journal of Advances in Modeling Earth Systems, 13(12), e2021MS002647. https://doi.org/10.1029/2021MS002647

---

## Author Response (AR1)

Comments on GMD-2024-94: Sunburned plankton: Ultraviolet radiation inhibition of phytoplankton photosynthesis in the Community Earth System Model version 2 Authors: Coupe et al.

General Comments – The authors have implemented for the first time an earth system model that includes the effects of solar UV irradiance on the photosynthesis of marine phytoplankton. Although there have been several modeling exercises that address the effect of UV inhibition of photosynthesis on, e.g., daily areal productivity for global or regional (e.g. Southern Ocean) basis, this is the first time those responses have been integrated into a full ecosystem model with the provision of feedback effects and shifts in taxonomic composition. This new model is potentially useful in inferring what effects UV currently has on the marine pelagic ecosystem as well as how these effects may change in response to various extreme events or climate manipulations. An additional use case would be to compare the model output for scenarios with and without the controls on ozone depleting substances imposed by the Montreal Protocol. Such global assessments of the "world avoided" have thus far only been conducted for carbon cycling in terrestrial ecosystems (Young et al. 2021).

While I applaud the work of the authors in structuring the model, the implementation of the biological weighting functions and penetration of weighted UV radiation has several deficiencies. I expect that the model can be corrected to address these problems (detail provided below), so that the modeling and assessment community can have a CESM2- UVphyto that is consistent with our current understanding of UV effects on phytoplankton.

Thank you for your detailed and careful review of our manuscript! In the revised manuscript, we have corrected the model to address the previous deficiencies in the biological weighting functions and the penetration of UV radiation. We include a detailed response to your comments below.

**Specific Comments**

1) A general point that should be made clear to any user of the model is that sensitivity to inhibition by UV irradiance is a physiological characteristic that is as variable as any other parameter of phytoplankton photosynthesis. Sensitivity is variable mainly because net inhibition reflects that balance between damage and repair processes (e.g. Neale and Kieber 2000). Variability on the damage side primarily derives from physical characteristics – e.g. the optical characteristics and cell dimension – these most often vary in a narrow range for any one taxa. However, repair processes can vary considerable depending on growth conditions. In this version of CESM2-UVphyto, Biological Weighting Functions (BWFs) are fixed irrespective of

growth conditions (except for CO2 – see below). This is an inconsistency in the code since the MARBL model does incorporate photoadaptation of photosynthesis in general through the C:Chl ratio, responding to growth irradiance, temperature and nutrients. It shouldn't be surprising that sensitivity to UV inhibition is also affected by these factors and there are many studies that confirm this beyond the studies cited for the model (see bibliography below). Where variation is known, e.g. for temperature and growth irradiance (cf. Neale and Thomas 2016), the code should account for it. If UV inhibition response is known only for one growth condition, the model should warn of increased uncertainty for predictions beyond the experimental conditions used for determination.

The biological weighting functions selected for UV inhibition in the model do not vary depending on temperature or PAR. Such complexity is not possible for all BWFs, as BWFs are often available at only one or two temperatures. In response to this comment, we have added text to the Discussion section of the manuscript explaining the limitations of the current version of the model and possible next steps to add more complexity.

"We caution that the model is somewhat sensitive to the exact BWF employed and because the PFTs in MARBL represent many different types of phytoplankton while the BWFs are based on single species, there is no "correct" BWF. Furthermore, there is a limited temperature range at which BWFs are reported, typically between 20C and 26C, affecting model performance at very cold temperatures."

2) The one condition for which the model varies sensitivity to UV is in relation to atmospheric CO2, which changes the pCO2 and pH of ocean water. Ironically, the BWFs for the microalga chosen to simulate the effect of changing CO2 show no evidence that they are affected by elevated CO2. Lorenzo et al. (2019) compared the BWFs of E. huxleyi grown at equilibrium with atmospheric CO2 of 400 and 800 ppm and found no difference even though there were changes in the coccoliths. Xu et al (2016) observed somewhat contrasting results in that a calcified strain was more resistant to UV than a naked strain. They concluded that coccoliths have an effect protecting against UV, however the experiment was not controlled in the sense that there were several differences between the + and – UV treatments, besides UV (including strain, PAR level, variability of exposure). These differences don't exclude that coccoliths perform a screening function, however the study of Lorenzo et al was performed under controlled conditions so that CO2 was the only factor that varied. In this case, the changes in coccoliths were insufficient to affect sensitivity to UV or if the change in coccoliths did allow more damaging UV to reach the cell, the effects were compensated by enhanced repair capabilities (this is discussed by Lorenzo et al). In summary, it would be inappropriate to vary the Einh (\* omitted for convenience) computed from the Lorenzo et al BWF according to the PIC/POC ratio when Lorenzo et al did not observe an effect. As an aside, other taxa do show increased sensitivity to UV under CO2 enhancement and it would be interesting to evaluate their responses in the context of the model (see Sobrino et al. In principle, I expect that the model code can be changed to accommodate this variability, which I encourage the authors to do. But I also recognize that the BWFs used in the case studies shown could be regarded as a "proof of concept" choices. This is defensible as along as it is made clear that results could be quite different for other choices, even for taxa within the same PFT group. However, independent of the choice of BWF, there are several steps described in the calculation of inhibition of photosynthesis section that are incorrect and lead to results that are inconsistent with current understanding of UV effects.

The species used to construct the coccolithophore PFT in MARBL is much more lightly calcified than that used by Lorenzo et al. in their 2019 study. While Lorenzo et al. (2019) did not observe a decrease in PIC/POC in response to increasing  $CO_2$  in their heavily calcified strain, experiments conducted with other species indicate a decline in PIC/POC with increasing CO2 which has consequences for UV sensitivity of coccolithophores (Xu et al, 2011; Guan et al., 2010). Thus, in the version of CESM2-UVphyto that we use for our simulations, we maintain the PIC/POC scaling for UV sensitivity. We note that this feature can be toggled on or off in the model for other users.

In response to this comment, we have included this information in the text:

L243: "For coccolithophores, we adopted the BWF reported for a heavily calcified species of Emiliana huxleyi"

and note that the scaling can be toggled on and off in L283.

"the species used by Lorenzo et al. (2019) is heavily calcified, unlike the species used to construct the coccolithophore PFT in MARBL. We include a scaling enhancement of UV inhibition as a function of coccolithophore shell thickness to the model which can be toggled on by the model user."

3) The propagation of Einh through the water column cannot be approximated with the attenuation of a single wavelength (Eqs 2 and 3). Although it is often used as a proxy for the attenuation of DNA damaging UV-B, the attenuation coefficient at 305 nm is inappropriate for propagating Einh(z) because most of the weight derives from UV-A. As a result, Einhz propagated with Kd305, declines with depth much faster than that calculated with a fully spectral resolved  $Kd(\lambda)$ , as shown in this example for clear oceanic water :

Figure 1: Depth profiles of Einh estimated using either a single attenuation coefficient (Kd(305)) applied to weighted irradiance at the surface (solid line) or using spectral attenuation coefficients (dashed line). Spectral irradiance and attenuation coefficients in the Pacific at 15°S at midday were estimated as described by Neale and Thomas (2016) and Einh calculated using a BWF for *Synechococcus* (ML@26°C, Neale et al. 2014). For this profile, Kd (305) was 0.139 m-1.

The depth to which UV inhibition affects photosynthesis in clear ocean waters is much deeper than 16 m (see also Fig. 2 in Neale and Thomas 2016). The effective Kd(z) (=-  $\ln(Einh(z+1)/Einh(z))$  in this example is similar to that of Kd( $\lambda$ =327nm) at the top of profile but changes (decreases) progressively to be similar to Kd( $\lambda$ =388 nm) at 100 m, with about 2.4x change in apparent Kd over the profile.

We thank the reviewer for this detailed analysis of the shortcomings of our current attenuation scheme. Ideally, we would have high spectral resolution of E\*inh for use in the oceans. However, because of computational resources there are limitations regarding the number of fields that can be passed from the atmosphere to the coupler and to the ocean at each timestep. It was this constraint that informed the initial design.

We have refined the spectral resolution of E\*inh by separating E\*inh into UV-A, UV-B and UV-C components. New attenuation coefficients for the different components of E\*inh are detailed in the response to Specific Comment #4.

4) Therefore, a more wavelength resolved approach is needed to propagate Einh. The values of Overmans and Agusti (2020) for coral reef areas in the Red Sea are inappropriate to apply over the whole ocean (Eq 2). Many areas of the ocean have more UV transparency than the Red Sea. Tedetti and Sempéré (2007, Table 2) reviewed global measurements of UV penetration and report that most open ocean waters have, e.g., 10%UV-B depths > 8 m. The maximum 10%UV-B depth possible from the Overmans and Agusti equation is 2.3/.29 =7.9 m. UV penetration is higher

in the open ocean because it is further from land and has lower concentrations of colored dissolved organic matter (CDOM) than the Red Sea. CDOM is more important in determining UV transparency than Chl. One possible approach to a more representative spectral Kd is to use the equation of Lee et al (2013) (see their Eq. 5), estimating the IOPs of absorption and backscattering based on MARBL parameters – mainly Chl, POC and DOC.

The chlorophyll approach was chosen because the existing formulation for visible light propagation in the model uses chlorophyll information. Because of the high correlation between dissolved organic matter and chlorophyll in the open oceans in our model, we maintain the use of chlorophyll but have abandoned the Overmans and Agusti (2020) approach. To better represent open ocean conditions, which comprises the majority of grid cells in our model, we constructed new attenuation coefficients based on the chlorophyll and Kd data in Tedetti et al. (2007) for UV-A and UV-B radiation. Because there is very little work studying UV-C radiation attenuation, Smith and Baker (1981) attenuation coefficients for clear seawater were used to construct the UV-C attenuation coefficients. A figure showing the Tedetti et al. (2007) data and regression lines is shown below.

**Regression of CHL onto Kd based on Tedetti et al. (2007)**

Attenuation coefficients (Kd) have now been changed to the following, where x is

```
chlorophyll and Kd is the attenuation coefficient: UV-A radiation, Kd(\lambda~340 nm): Kd = 0.243728x + 0.041839 UV-B radiation, Kd(\lambda~305 nm): Kd = 0.437891x + 0.129573 UV-C radiation, Kd(\lambda~265 nm): Kd = 0.6x + 0.4
```

Using Smith and Baker (1981) values for pure seawater as a reference, the attenuation coefficient at the selected wavelengths are now similar for the minimum chlorophyll value in MARBL (0.02 mg m-3). Only Kd(340 nm) has slightly lower attenuation than the values determined in Smith and Baker (1981). However, because of the large weight of wavelengths above 340 nm, Kd(340) is likely closer to Kd(340 to 360 nm). Below are the Smith and Baker (1981) values, as a comparison: Kd(340 nm)=0.0637 Kd(305 nm)=0.12 Kd(265 nm)=0.4

5) BWFs in the literature are part of specific integrated photosynthesis-irradiance relationships (BWF/P-I models), and only correctly estimate UV inhibition if implemented using the relationship for which they were defined. There are three BWF/P-I models: E, T and Emax (see comparative discussion in Lorenzo et al. (2019), and each BWF used in the CESM2-UVphyto case studies uses a different linked model. Particularly important is the difference between the E model used for the Diatom (*Phaeodactylum*) BWF (1/[1+Einh] dependence) and the T model used for the Coccolithophore (*E. huxleyi*) BWF (1/Einh dependence). Evaluating UV inhibition by substituting the *E. huxleyi* Einh into the E model predicts more inhibition than if implemented with the correct T model (Fig. 2)

inhibition if the models are different.

Figure 2. Examples of the depth profiles of yuv evaluated with three different BWF models using spectral irradiance in the Pacific at 15°S at midday (cf. Fig 1). (1) Syn-Uses a BWF for Synechococcus (ML@26°C. Neale et al. 2014) using the same Einh in either the Emax (solid green) or E model (dashed green); (2) Phaeo-Phaeodactylum sp. using the E model (brown line) and (3) Ehux - Emiliana huxlevi using the T model (black solid) or same Einh in the E model (dashed black line).

For the T model prediction of *E. huxley*i inhibition (Fig. 2), there is a sharp decline in inhibition with depth as this model was optimized to best predict responses at high exposure. The same Einh evaluated with the E model has lower  $\gamma_{UV}$  (more inhibition) and penetrates deeper. Lorenzo et al acknowledged that there probably is some inhibition at lower exposures (and deeper in the water column), but this could not be resolved given the variability of the laboratory measurements, so below a certain depth (23 m in the above) there is no effect. In the example, the overall inhibition of midday productivity over the upper 100 m, given the P-I parameters in Lorenzo et al, is only 11% with the T model, but increases to 30% if the T model Einh is used to predict  $\gamma_{UV}$  using Equation 4 (E model). This directly impacts conclusions drawn from the reported case studies that coccolithophores are the most sensitive to inhibition by UVR. Higher Einh, per se, doesn't mean more

We thank the reviewer for pointing out this crucial mistake in the model. We now have updated the model using E, T, or Emax when appropriate. This, in conjunction with altering the attenuation coefficients, has slightly modified the results. A new figure has been added that relates E\*inh and  $\gamma_{UV}$  for all of the different models employed, as a subpanel in Figure 1.

New version of Figure 1.

6) On the other hand, for the small prokaryotes, *Prochlorococcus* and *Synechococcus*, response at low exposure (which varies as 1/[1+Einh]) could be resolved and distinguished from that at high exposure (1/Einh dependence), the two responses are combined in the Emax model (Neale et al 2014). The BWF/PI for both Pro and Syn use the Emax model, which in this case leads to more inhibition for the Einh(PI) case compared to using the 1/[1+Einh] form over the whole water column (Fig. 2). Use of the correct model combined with more accurate Einh propagation should result in much higher relative effect of UV on ocean productivity for Einh(PI) than the ~1% effect in the reported case studies. The cited range of 7-28% inhibition reported by Neale and Thomas 2016 is the inhibition of integrated midday production by UV (vs only PAR) in their simulations of the Pacific for current conditions, not what was observed in the laboratory. Similarly, estimates of around 7%

inhibition of daily integrated production by UVR have been obtained by other approaches, e.g. Cullen et al. (2012) and Moreau et al. (2015).

We employed the updated BWF models with attenuation at a higher spectral resolution, as suggested, and found an increase from the 1% reported effects, especially in the surface oceans. The manuscript has been updated to reflect the new results.

7) The BWF function chosen for small phytoplankton is for *Prochlorococcus*, which is much more sensitive to inhibition by UV radiation than other picophytoplankton (Neale and Thomas 2016). It is better consider Pro a separate case. The BWFs for *Synechococcus* is probably more representative of picophytoplankton overall as well as diazotrophs.

We thank the reviewer for this suggestion. We have adopted the new *Synechococcus* BWF using the Emax model from Neale and Thomas (2014) in the new version of the model. This can be seen in the new Figure 1.

8) Equation 1 should include a term for inhibition by PAR (εPAR\*EPAR cf. Eq. 3 in Neale and Thomas 2016). PAR inhibition is significant for the cases of *E. huxleyi*, *Prochlorococcus* and *Synechococcus*, and other published BWFs. No PAR inhibition was defined for *Phaeodactylum*, that is partly related to the lower EPAR exposures used in this first experimental determination of the BWF.

We thank the reviewer for bringing the importance of PAR inhibition to our attention. We examined the PAR fields in our simulations and do not find sufficient conditions for significant PAR inhibition. The existing MARBL photosynthesis calculation is already intended to capture phytoplankton behavior at higher PAR levels. We have added a sentence that makes mention of PAR inhibition, but we clarify that it will not be a significant factor in the pre-industrial simulations presented in this manuscript.

L195: "While some laboratory studies include a PAR inhibition term in E\*inh (Neale and Thomas, 2016), our model configuration does not account for this term; in our simulations, high UV is not accompanied by high PAR."

9) Elevated UV radiation. Because most inhibition is caused by UV-A radiation, Einh at the surface for known BWFs will never be increased by a factor of 20 vs Einh(PI), even if 95% of stratospheric ozone is destroyed (total ozone column would be reduced by a lower fraction as the increased penetration of UV-C will form ozone at lower altitudes). Thomas et al. (2015) treated the case of a gamma-ray burst resulting, briefly (months), in a strong depletion of column ozone by 70% in the region of the strike. At most, this increased the Einh(0) by a factor of 2.63, based on the Phaeo BWF which has probably the largest ratio of sensitivity to UV-B vs UV-A. The multiplier will be less for almost all other phytoplankton especially when PAR inhibition is included. The effect of a gamma-ray burst on Integrated production at midday was also evaluated for *Synechococcus* and *Prochlorococcus* (Neale and Thomas 2016b) and ozone depletion of 70% resulted in at most 3% additional inhibition (vs normal ozone) to the 1% depth of PAR and 7% additional inhibition of productivity integrated over the mixed layer.

We agree that the E\*inh(20x PI) values are overly simplified and may produce unrealistic E\*inh values. Instead of this simplified approach employed in the

previous version of the manuscript, we re-ran the model with a fully coupled high-top atmospheric model with full stratospheric chemistry. The new simulation included halogen amounts comparable to the Chicxulub asteroid impact, 117,000 Tg, as estimated by Toon et al. (2016). We now include this simulation in Materials and Methods with a forcing that is defined as E\*inh(halogen) for the ocean-only simulations. However, we find that the E\*inh values for these simulations are even greater than 20x E\*inh(PI) for small phytoplankton and diatoms, but not coccolithophores. Increased UV-B radiation, especially in the wavelength range where the BWF curves increase exponentially as a function of decreasing wavelength, is responsible for this. In this case, total column ozone is depleted by more than 95%. This is an extreme upper bound test case, where even UV-C radiation reaches the surface. We intend to use this model to test UV radiation after the K-Pg boundary in future work. In response to this comment the manuscript now reads:

L310: "We conduct 5 year simulations to explore the modeled biogeochemical and ecological response to extremely high levels of surface UV radiation. Halogens equivalent in quantity to the Chicxulub asteroid impact at the K-Pg boundary (Toon et al., 2016) are injected into the stratosphere. The halogen injection includes hydrogen chloride and hydrogen bromide and is intended to mimic an upper bound of a possible surface UV radiation perturbation. At the same time, the halogens are unlikely to block visible or ultraviolet radiation from reaching the surface and will minimize changes to other aspects of the climate, circulation feedbacks in response to depleted ozone notwithstanding. This case is referred to as E\* inh(halogen). A fully coupled simulation is run for two years and coupler forcing is used to generate a five-year offline simulation; for simplicity, years 3-5 of the offline simulation are repeated versions of year 2 forcing."

10) Yet another approach is needed for polar phytoplankton, for which low light and temperature can result in very low repair rates such that in some cases (mainly deeply mixed zones), inhibition is time dependent (see Neale et al 1998, Smyth et al 2012). Probably polar oceans, especially S. Ocean inside ice-limit, should be treated as a special case.

We agree that treating polar phytoplankton separately would be the most accurate way to simulate their response to changes in both PAR and UV radiation. MARBL, in its current configuration, is not suited to treat polar phytoplankton separately or to include a time dependent inhibition. This would likely require adding a fifth phytoplankton functional type. Text has been added to the revised manuscript that addresses the simplified nature of our model:

L146: "Because of the diversity of the phytoplankton contained within the small phytoplankton functional group, phytoplankton in high latitude regions that are often highly temperature and light limited may not be as well represented in this model."

11) Finally, the CESM2-UVphyto model assumes that the only effect of UV on phytoplankton is through inhibition of photosynthesis. However, UV (more specifically UV-B) also directly damages DNA, inhibiting growth. Too little is known to quantify the importance of this mode of UV effect on a global basis, still it should be recognized that it could be important (see Andreasson and Wangberg 2007).

We have made sure to include in the text that UV can directly damage DNA and cause long-term effects that are not represented. This can be found in "Discussion":

L442: "BWFs are typically determined from short-term growth inhibition, which may not reflect the effects of direct damage to DNA over longer timescales"

Summary – The model needs extensive revision after which the case studies can be re-run. At that point the results and conclusions can be re-examined and re-written as needed. Detailed reviews of those sections will be provided then. Minor and Technical Comments

Line 95 Although stratification has intensified, surface mixed layer depths are not increasing, contrary to the expectations of Gao et al. (2019). See discussion in Neale et al. (2023)

We have modified this discussion on global warming, stratification, and UV exposure to include nuance about present and future trends in mixed layer depths.

L94: "Finally, marine phytoplankton exposure to UV radiation **may increase** in some regions as anthropogenic climate change warms the Earth's surface, representing a compounding threat. The warming of the Earth's surface in regions where wind speeds do not increase may increase the density gradient in the upper ocean."

Figure 1 The text in several places states limits to UV-B as 280-315 nm, but this plot has the division between the bands at 320 nm

Figure 1 has been modified so that the bands are instead at 315 nm.

Line 230 Supplemental Table 1 only lists weights up to 327 nm

Supplemental Table 1 has been extended to the full wavelengths.

Line 230 Relative to the originally published BWFs, weights have been both interpolated and extrapolated.

BWFs need to be interpolated to the atmosphere model grid to use the weights. This has been clarified in the text. Extrapolation to UV-C wavelengths was necessary because they were not reported in all of the published BWFs, yet based on preliminary simulations of the K-Pg impact, UV-C radiation may be important for UV damage. We used information from the literature about the one BWF that did report UV-C damage information to extrapolate. This information has been added to the methods.

L240: "Wavelengths are interpolated to the bounds provided in this table to calculate spectral integrals"

L266: "Not all of the employed BWFs extend into wavelengths below 280 nm (UV-C radiation). While UV-C radiation is a non-factor in recent history, it may become relevant after a cataclysmic asteroid impact. To account for UV-C radiation damage, BWFs are extrapolated to 200 nm, as indicated by dashed lines in Figure 1a."

Line 265 As mentioned above, Lorenzo et al. 2019 did not find an effect of PIC/POC ratio on the BWF. In any case, the treatment effect found by Xu et al (which could be UV as well as other factors) is more accurately defined as 3.5/2=1.75, since the calcified strain grew twice as fast as the naked strain under PAR-only incubator conditions.

We have modified the text to clarify this as an optional feature which can be turned on and off. In the text, we elaborate on the effects of having this PIC/POC scaling feature.

As suggested, we modified the PIC/POC scaling factor from 3.5 to 1.75 to isolate UV effects only (and not PAR effects). As a result of this modification, coccolithophore decline in response to UV radiation has been moderated.

Line 520 Arrigo 1994 – journal name missing

The journal name has been updated.

Line 620 Neale and Thomas GCB, publication year is 2016 The year has been updated to 2016.

Respectfully submitted,

Patrick Neale

**References:**

- Andreasson, K. I. M., & Wängberg, S.-A. (2007). Reduction in growth rate in *Phaeodactylum tricornutum* (Bacillariophyceae) and *Dunaliella tertiolecta* (Chlorophyceae) induced by UV-B radiation. Journal of Photochemistry and Photobiology B: Biology, 86(3), 227-233.
- Banaszak, A. T., & Neale, P. J. (2001). UV Sensitivity of photosynthesis in phytoplankton from an estuarine environment. Limnol Oceanogr, 46, 592-600.
- Cullen, J. J., R. F. Davis, and Y. Huot (2012), Spectral model of depth-integrated water column photosynthesis and its inhibition by ultraviolet radiation, Global Biogeochem. Cycles, 26, GB1011, doi:10.1029/2010GB003914.
- Lee, Z., C. Hu, S. Shang, K. Du, M. Lewis, R. Arnone, and R. Brewin (2013), Penetration of UV-visible solar radiation in the global oceans: Insights from ocean color remote sensing, J. Geophys. Res. Oceans, 118, doi:10.1002/jgrc.20308.
- Litchman, E., & Neale, P. J. (2005). UV ECects on photosynthesis, growth and acclimation of an estuarine diatom and cryptomonad. Mar Ecol Prog Ser, 300, 53–62. Litchman, E., Neale, P. J., & Banaszak, A. T. (2002). Increased sensitivity to ultraviolet radiation in nitrogen-limited dinoflagellates: photoprotection and repair. Limnol Oceanogr, 47, 86-94.
- Moreau, S., Mostajir, B., Bélanger, S., Schloss, I. R., Vancoppenolle, M., Demers, S., & Ferreyra, G. A. (2015). Climate change enhances primary production in the western Antarctic Peninsula. Global Change Biology, 21(6), 2191-2205. https://doi.org/10.1111/qcb.12878
- Neale, P. J., Banaszak, A. T., & Jarriel, C. R. (1998). Ultraviolet sunscreens in dinoflagellates: Mycosporine-like amino acids protect against inhibition of photosynthesis. J Phycology, 34, 928-938.
- Neale, P. J., Cullen, J. J., & Davis, R. F. (1998). Inhibition of marine photosynthesis by ultraviolet radiation: Variable sensitivity of phytoplankton in the Weddell-Scotia Sea during the austral spring. Limnol Oceanogr, 43, 433-448.
- Neale, P. J., & Kieber, D. J. (2000). Assessing biological and chemical effects of UV in the marine environment: Spectral weighting functions. In R. E. Hester, & R. M. Harrison (Eds.), Causes and Environmental Implications of Increased UV-B Radiation, 14, (pp. 61-83). Royal Society of Chemistry
- Neale, P. J., Pritchard, A. L., & Ihnacik, R. (2014). UV effects on the primary productivity of picophytoplankton: biological weighting functions and exposure response curves of Synechococcus. Biogeosciences, 11, 2883-2895. <a href="https://doi.org/10.5194/bqd-10-19449-2013">https://doi.org/10.5194/bqd-10-19449-2013</a>
- Neale, P. J., & Thomas, B. C. (2016b). Solar irradiance changes and phytoplankton productivity in Earth's ocean following astrophysical

- ionizing radiation events. Astrobiology, 16(4), 245-258. https://doi.org/10.1089/ast.2015.1360
- Neale, P. J., Williamson, C. E., Banaszak, A. T., Häder, D. P., Hylander, S., Ossola, R., Rose, K. C., Wängberg, S. Å., & Zepp, R. (2023). The response of aquatic ecosystems to the interactive effects of stratospheric ozone depletion, UV radiation, and climate change. Photochemical & Photobiological Sciences, 22(5), 1093-1127.
  - https://doi.org/10.1007/s43630-023-00370-z
- Smyth, R. L., Sobrino, C., Phillips-Kress, J., Kim, H.-C., & Neale, P. J. (2012). Phytoplankton photosynthetic response to solar ultraviolet irradiance in the Ross Sea Polynya: Development and evaluation of a time-dependent model with limited repair. Limnol Oceanogr, 57(6), 1602–1618.
- Sobrino, C., & Neale, P. J. (2007). Short-term and long-term effects of temperature on phytoplankton photosynthesis under UVR exposures. J Phycol, 43, 426-436. Sobrino, C., Neale, P. J., & Lubian, L. (2005). Interaction of UV-Radiation and Inorganic Carbon Supply in the Inhibition of Photosynthesis: Spectral and Temporal Responses of Two Marine Picoplankters. Photochem Photobiol, 81, 384–393.
- Sobrino, C., Ward, M. L., & Neale, P. J. (2008). Acclimation to elevated carbon dioxide and ultraviolet radiation in the diatom Thalassiosira pseudonana: Effects on growth, photosynthesis, and spectral sensitivity of photoinhibition. Limnol Oceanogr, 53, 494-505.
- Tedetti, M., & Sempéré, R. (2006). Penetration of Ultraviolet Radiation in the Marine Environment. A Review. Photochemistry and Photobiology, 82(2), 389-397. <a href="https://doi.org/http://dx.doi.org/10.1562/2005-11-09-IR-733">https://doi.org/http://dx.doi.org/10.1562/2005-11-09-IR-733</a>
- Young, Paul J.; Harper, Anna B.; Huntingford, Chris; Paul, Nigel D.; Morgenstern, Olaf; Newman, Paul A.; Oman, Luke D.; Madronich, Sasha; Garcia, Rolando R.. 2021. The Montreal Protocol protects the terrestrial carbon sink. Nature, 596 (7872). 384-388. https://doi.org/10.1038/s41586-021-03737-3

**References for review:**

Smith, R. C., & Baker, K. S. (1981). Optical properties of the clearest natural waters (200–800 nm). Applied Optics, 20(2), 177–184. https://doi.org/10.1364/AO.20.000177

|    |    |     |     |    | === |
|----|----|-----|-----|----|-----|
| RE | ΞV | IΕ\ | NΕ  | R  | 2   |
| == |    | === | === | == | -== |

Sunburned plankton: Ultraviolet radiation inhibition of phytoplankton photosynthesis in the Community Earth System Model version 2, by Joshua Coupe et al.

This paper presents modifications made to CESM2 to enable the simulation of UV-B (280- 315 nm) and UV-A (315-400 nm) propagation through the ocean and UV inhibition of phytoplankton photosynthesis. Biological weighting functions (BWF) that determine UV inhibition were incorporated into the Tropospheric Ultraviolet and Visible (TUV) model, in order to calculate spectrally-integrated BWFs at an hourly frequency. The weighting functions of inhibition were then applied in the ocean biogeochemistry model MARBL, which represents 4 phytoplankton PFTs – small phytoplankton, diatoms, diazotrophs, and coccolithophores; and 2 zooplankton PFTs. In MARBL, coccolithophore shell thickness can be impacted by CO2 levels.

The damage functions were based on the few available studies of productivity responses to changes in UV radiation. There are several pertinent applications and needs for Earth system models to represent these impacts, and this study is an important step forward for the CESM community and would be of interest to other modelling groups. The paper describes the modelling approach clearly and the results are described with an appropriate level of detail. My main comments relate to some inconsistencies in the description of the study and results, organization, and framing of the results.

**Major comments:**

- 1. Scale of UV changes:
- a. How do the PI UV radiation levels compare to present-day? Since occasionally the results are compared to existing studies, it would be helpful to add a discussion of the differences between present-day and preindustrial UV, and how these might impact the results.

We have added a section to the manuscript that puts UV radiation levels in the pre-industrial into context with present day levels. This is now in the Materials and Methods section that describes the pre-industrial simulations:

L305: "Because of the lack of ODS [ozone depleting substances] in the pre-industrial stratosphere, UV radiation is elevated in the present day compared to the pre-industrial in terms of UV index by up to 13% in tropical regions, 10% in the Antarctic, and 4% in Arctic regions."

Furthermore, we have added to the discussion regarding these differences.

L4421: "The simulations presented here use pre-industrial boundary conditions and therefore exhibit slightly less UV radiation compared to similar simulations with present day ozone distributions. The phytoplankton response in the surface ocean is likely to be enhanced in the present day compared to the pre-industrial, but would still be less than the E\*inh(halogen) simulation."

b. 20xPI sensitivity study: This is a huge impact (95% global reduction in stratospheric ozone), if Pinatubo reduced NH ozone by 10% and Hunga Tonga reduced tropical ozone by

5% (also see Fleming et al. 2024 for more recent estimate of Hunga Tonga impacts on ozone from the water vapor injected into the stratosphere). I suggest returning to these discrepancies in the discussion, to highlight that the changes discussed would not be expected on a global scale (perhaps unless a very extreme case like asteroid impact or nuclear war), but instead they display the model sensitivities to changes in UV radiation. c. The discussion in the supplement of ozone hole stress would fit well into the main discussion of the manuscript and further helps put these results into context of historical impacts.

After considering the first review, we removed the 20x PI simulation for being unphysical and overly simplified. Additionally, E\*inh values for the different phytoplankton functional types would not each increase by 20x PI because of the way that the curves are designed. Instead, to test this upper bound, we conduct a halogen injection similar to the Chicxulub asteroid impact to test the most extreme UV radiation values globally. A description of this can be found in Section 2.6: Pre-industrial simulations with elevated UV radiation. We have also added to the discussion in the main text to include information about ozone hole stress.

2. Further discussion of the attenuation of radiation with depth: Equations 2-3 and the values of K imply that most UV radiation attenuates by 16 m with low chlorophyll concentrations. But Figure 5 shows changes down to the 30-40m layer (Layer 4).

In response to another reviewer, the scheme to estimate attenuation of UV radiation with depth has been completely changed (equations 2 through 4 in the updated manuscript). The previous method treated all UV radiation equally, when in reality, UV-A radiation would penetrate far deeper than UV-B radiation. To address this, there are different attenuation coefficients for UV-A, UV-B, and UV-C radiation employed in the model.

Does the attenuation of PAR with depth follow a different trajectory? It seems to me that if <1% of surface PAR reached below 16m, there would be very minor impacts in the 3rd and 4th layers. What are the changes to PAR at depth in these experiments? Has there been any validation of depth profiles of PAR vs biomass at these depths?

Yes, the attenuation of PAR follows a different trajectory. PAR considers phytoplankton biomass in its attenuation scheme and is able to penetrate further when the phytoplankton near the surface is removed, especially relative to UV radiation. Because the PAR extinction coefficient is orders of magnitude smaller than the coefficient for UV-B, this allows its 1% depth level to be up to 150 meters for lower chlorophyll concentrations. This is why there are increases in productivity down to 140 meters in the simulations with E\*inh(PI) UV radiation. We have added a few sentences to the Discussion that the low vertical resolution of the model may affect the simulation of vertical productivity.

L456: "Finally, because UV attenuates so quickly with depth, the available 10 m vertical spacing in CESM2-UVphyto may produce small inaccuracies in UV inhibition of photosynthesis that can affect the vertical profile of phytoplankton and as a result, the shading of PAR and PAR amounts deeper in the water column. Implementation of UV inhibition in a model with higher vertical resolution would likely resolve these processes with greater accuracy."

(Minor point: Line 374: This states there is an increase in NPP in the 3rd layer (25m), but Figure 5 shows the 4th layer (35m) – it would be better if the text and figures referred to the same

The figure showing productivity changes in the subsurface has been modified to show both the 1st layer (z = 5 m) and the 9th model layer (z = 85 m) and there is now text that references the depth specifically shown in Figure 5.

L380: "Deeper levels of the ocean experience a surge in productivity in response to increasing UV radiation (e.g., an 8% increase in NPP at 95 m)."

3. As this is a first step in modelling these impacts, I think the authors could draw even more attention in the discussion to the additional modelling and observational studies that would help inform future developments or address any of the limitations in this study. Consider devoting a short sub-section or paragraph in the Discussion to future research priorities to help progress this work.

Thank you for this suggestion. We have added to part of the Discussion section that emphasized how future work and how observational studies could improve modelling efforts.

L445: "Further research developing BWFs with laboratory studies that are more tailored to the species used to represent each PFT in MARBL and across larger temperature ranges would narrow uncertainties in simulating UV inhibition of photosynthesis. CESM2-UVphyto can be used to quickly assess how new laboratory-derived BWFs for different phytoplankton species would affect phytoplankton biomass, carbon export, and effects for marine organisms at upper trophic levels that are supported by phytoplankton."

4. Description of methods: The abstract states that you conducted simulations to calibrate estimates of sensitivity of phytoplankton productivity to UV radiation. But this is not reflected in the paper, since the experiments discussed only used one value of the biological weighting functions. It would be more accurate to say you did these experiments to 'understand' the sensitivity, unless some calibration was done that is not discussed in the paper (in which case, that should be explained in the manuscript). Also Line 485: I do not see evidence of a large parameter space of E\*\_inh being investigated in this study.

We removed "calibrate" and changed the abstract to "understand".

To avoid exaggeration with the "large parameter space" phrasing since we only explored E\*inh derived from two different stratospheric ozone states, we have changed the sentence to:

L467: "We explored global E\*inh values ranging from a healthy, pre-industrial stratosphere to a very depleted stratosphere to understand the performance of our modifications at extremes"

**Minor points**

Organization: The results seem out of order to me. Figure 4 is helpful for understanding where the different PFTs live, and the absolute impacts of UV radiation on their NPP. This helps with the understanding of the globally averaged results shown in Figure 2. I suggest rearranging and adding subsections, ie: (1) regional impacts of UV radiation on phytoplankton NPP; (2)

globally integrated impacts of UV radiation; (3) vertical distribution of impacts; (4) effect of enhanced CO2.

Thank you for the suggestion. After some attempts at rearranging, we decided that the manuscript read best when introducing the total globally averaged results first as the main metric by which we understand how UV forcing impacts global phytoplankton. This is similar to some of the other modelling literature involving marine ecosystems (see: Lovenduski et al., 2016, Krumhardt et al., 2017, Lovenduski et al., 2020, Fay et al., 2023).

Lines 151-154: What is the motivation for the sensitivity studies with different configurations of MARBL? They are not discussed in the manuscript (with exception of a very small mention of the 3p1z experiment in the results). So either remove the mention of these additional experiments, or update manuscript to include these in your results/discussion and more fully explain how they contributed to the understanding of phytoplankton sensitivities to UV radiation. (Moving the supplement into the main text would help, as this is relatively short and very relevant to the validation of your results)

We have removed the mention of the sensitivity tests from the main manuscript. At the beginning of the process of using this model and writing up the manuscript, the 4p2z configuration of MARBL was mostly untested and had not been used in a single scientific publication. The need to validate this version of the model is no longer necessary.

Equation 1: Over what interval of lambda are these calculated? (What is Δlambda?)

A description of the radiation bounds to compute E\*inh for UV-A, UV-B, and UV-C radiation has been added to the model components section.

L131: "UV-A radiation is between 320 nm and 400 nm, UV-B radiation is between 280 nm and 320 nm, and UV-C radiation is defined as 100 nm to 280 nm but the model is only able to compute spectral integrals from 121 to 280 nm."

Delta Lambda is defined in the Calculation of Ultraviolet Inhibition of Photosynthesis section and we have added a sentence that clarifies that lambda is between 121 nm and 400 nm but is separately calculated for UV-A, UV-B, and UV-C radiation.

L193: "[lambda] spans the UV spectrum of radiation from 121 nm to 400 nm and is subdivided into UV-A, UV-B, and UV-C components."

Line 333: Increase in productivity – is this in relative or absolute terms? Only relative terms are shown in Figure 2.

The line has been rephrased to:

L346: "The phytoplankton productivity response to UV inhibition is a function of characteristics unique to each phytoplankton type, with some types benefiting at the expense of others"

Figure 3: How is the total E inh calculated, is it weighted by PFT distribution?

The E\*inh figure (Figure 3 still) has been removed and replaced with gamma\_UV which can more directly be used to compute the effect on phytoplankton photosynthesis. The original figure did not weigh by PFT distribution. The new one is computed, weighing gamma\_UV by the gridcell productivity of each PFT. gamme\_UV is similarly affected by variations in latitude. A description of how this is computed has been added to the figure caption.

Figure 3 caption: "The spatial distribution of total average surface  $\gamma UV$  for all PFTs under (left) E \* inh(PI) and (right) E \* inh(halogen) over five years of simulation, weighted by the distribution of each PFT. Maximum annual sea ice extent is indicated by the blue solid line. Lower values of  $\gamma UV$  indicate greater plankton limitation"

Lines 355-360: Here, and elsewhere in the results, I would suggest you keep the description of the patterns and other direct results from these experiments, but leave the comparison to other studies for the Discussion.

Thanks for the suggestion. The validation of CESM2-UVphyto with existing versions of the model walks a fine line between results/discussion. We believe the paper reads best if we introduce the results with the validation of the model with respect to previous versions.

Lines 453ish: Mention here the amount of increased UV radiation in this region in these experiments compared to ozone hole or other historical events.

We removed this section from the discussion. However, we have now added information about the amount of ozone decline and UV index in the E\*inh(halogen) simulations compared to what was recorded over Antarctica during the ozone hole.

L344: "Under E\*inh(halogen) stratospheric ozone declines by more than 90%, producing UV indices greater than 20, exceeding the greatest values of up to 14 over Antarctica under the ozone hole."

Lines 458-459: Provide a reference here and clarify – this is not clear – the responses are on the high end of observed values of productivity or observed values of productivity changes?

We removed this sentence completely from the text.

Line 465ish: In addition, I would think that phytoplankton are adapted to ambient UV levels

(this is the case for terrestrial plants), which would have an impact on the spatial variation of the inhibition factor / BWFs. Would it be important to consider the ambient levels of UV to which phytoplankton are accustomed, before considering impacts of changes to those levels?

Some work on phytoplankton in the Southern Ocean suggest a decline in productivity under even ambient levels of UV radiation (350 DU ozone layer). We moved a validation section from the Supplemental Information document to the Discussion that includes references to this work and puts declines under the ozone hole in context. See section starting at L424.

"Observational studies of regional phytoplankton productivity changes in response to ambient UV radiation range from a 0.15% annual mean reduction in NPP south of the Polar Front in the Southern Ocean (Helbling et al., 1992) to a 4% to 7% reduction in NPP during austral spring across the Southern Ocean (Prézelin et al., 1994). Smith et al. (1992) found a 3% reduction in a population of Phaeocystis when exposed to typical UV radiation levels, equivalent to an ozone layer with a thickness of 350 dobson units (DU), which is higher than typical values simulated under E\* inh(PI) (310 DU). In the E\* inh(PI) simulation, some parts of the Southern Ocean experience up to a 15% decline in NPP, adjacent to areas with an equally largeincrease in productivity. On average the Southern Ocean experiences a 5% to 10% decline in annual mean productivity, driven by coccolithophore decline. Small phytoplankton, the PFT most closely resembling Phaeocystis, experience a 3% decline at most and benefit in areas with the greatest coccolithophore loss. This is on the lower end of the Smith et al. (1992)'s findings for Phaeocystis but precise validation is made difficult by the large uncertainty range. The primary difficulty lies in the lack of information regarding how phytoplankton across most of the world, which did not experience an ozone hole, would beimpacted by increased UV radiation."

Lines 425-427: are you sure it's this way around? Could it be that the decreases in small phyto and diatoms are making it easier for coccolithophores to live here?

With the development of the new BWFs and attenuation scheme, these results have been modified and this statement has been removed from the text. The correlation is less clear in the new results. The new version of this text reads as follows:

L402: "In general, coccolithophores tend to be most negatively impacted by UV radiation in the seasonally ice-covered and subpolar biomes of the north Pacific, north Atlantic, and Southern Ocean (Figure 8a). The abundance of coccolithophores in these biomes, coupled with their relatively low PIC/POC values mean that coccolithophores are responsive to UV radiation increases here even under low levels of atmospheric CO2. In contrast, coccolithophore NPP increases in response to UV radiation in subtropical and tropical biomes under low levels of atmospheric CO2, as small phytoplankton productivity declines (Figure 8a). Under atmospheric CO2 of 900 ppm, most biomes show a loss in coccolithophore productivity with increasing UV radiation (Figure 8b). Figure 8c illustrates the role of increasing CO2 on coccolithophore NPP sensitivity to UV radiation in the Southern Ocean Subtropical Seasonally Stratified biome (SO-STSS; biome 15). Here, coccolithophore NPP reductions under UV radiation are enhanced by 5-30% when atmospheric CO2 increases from 284 ppm to 900 ppm, with the largest enhancements in April, November and December (Figure 8c)."

Line 436-437 – provide a reference here, but also consider moving comparisons of results to other studies to the discussion.

We removed this comparison completely because there wasn't a sufficient reference, especially in the context of the new simulation results.

Paragraph from Lines 428-438: When discussing PIC/POC scaling, is this for the case of 900 ppm? This is implied since it references Figure 8c. Please clarify.

Initially, the analysis lines 428 to 438 was only under 284 ppm and the Figure 8c reference was a mistake. We have significantly condensed this analysis and the biome figure (still Figure 8) highlights the effect of CO2 instead of showing the elevated UV simulations.

**References**

Fleming, E. L., Newman, P. A., Liang, Q., & Oman, L. D. (2024). Stratospheric temperature and ozone impacts of the Hunga Tonga-Hunga Ha'apai water vapor injection. Journal of Geophysical Research: Atmospheres, 129, e2023JD039298. https://doi.org/10.1029/2023JD039298